# Single-site pyrrolic-nitrogen-doped $sp^2$-hybridized carbon materials and their pseudocapacitance

Kesong Tian [1,2,3], Junyan Wang [3], Ling Cao [3], Wei Yang [3], Wanchun Guo [1,3 ✉], Shuhu Liu [4], Wei Li [2], Fengyan Wang [3], Xueai Li [1,3], Zhaopeng Xu [1], Zhenbo Wang [5], Haiyan Wang [1,3 ✉] & Yanglong Hou [2 ✉]

Integrating nitrogen species into $sp^2$-hybridized carbon materials has proved an efficient means to improve their electrochemical performance. Nevertheless, an inevitable mixture of nitrogen species in carbon materials, due to the uncontrolled conversion among different nitrogen configurations involved in synthesizing nitrogen-doped carbon materials, largely retards the precise identification of electrochemically active nitrogen configurations for specific reactions. Here, we report the preparation of single pyrrolic N-doped carbon materials (SPNCMs) with a tunable nitrogen content from 0 to 4.22 at.% based on a strategy of low-temperature dehalogenation-induced and subsequent alkaline-activated pyrolysis of 3-halogenated phenol-3-aminophenol-formaldehyde (X-APF) co-condensed resins. Additionally, considering that the pseudocapacitance of SPNCMs is positively dependent on the pyrrolic nitrogen content, it could be inferred that pyrrolic nitrogen species are highly active pseudocapacitive sites for nitrogen-doped carbon materials. This work gives an ideal model for understanding the contribution of pyrrolic nitrogen species in N-doped carbon materials.

[1] State Key Laboratory of Metastable Materials Science and Technology, Yanshan University, 066004 Qinhuangdao, China. [2] Beijing Innovation Center for Engineering Science and Advanced Technology (BIC-ESAT), College of Engineering, Peking University, 100871 Beijing, China. [3] Hebei Key Laboratory of Applied Chemistry, College of Environmental and Chemical Engineering, Yanshan University, 066004 Qinhuangdao, China. [4] Institute of High Energy Physics, Chinese Academy of Sciences, 100049 Beijing, China. [5] School of Chemistry and Chemical Engineering, Harbin Institute of Technology, 150080 Harbin, China. ✉email: wc-g@ysu.edu.cn; hywang@ysu.edu.cn; hou@pku.edu.cn

Carbon-based nanomaterials exhibit versatile flexibility in their fundamental properties, such as their mechanical, thermal, optical, and electrical properties, which are dependent on the hybridization of carbon, including $sp^3$, $sp^2$, and $sp^1$[1–8]. Doping heterogenous atoms, including nonmetal atoms (e.g. B, N, O, S, and P)[9–11] and metal atoms (e.g. Pt, Fe, Co, Ni, Zn, and Mo)[12–16], into the framework of carbon-based nano materials could further tailor their fine electron structure to enhance their performance in the field of energy storage and conversion. In particular, facilely doping nitrogen species into $sp^2$-hybridized carbon materials has been proved to be efficient to improve their performance[17–19] in electrochemical fields, such as pseudocapacitance[20,21], oxygen reduction reaction (ORR)[22–25], $CO_2$ reduction reaction[26–28], oxygen evolution reaction (OER)[29–31], hydrogen evolution reaction (HER)[32], nitrogen reduction reaction[33–35], and so on, mainly because nitrogen doping could tailor local electron structures and/or induce substitutional defects of carbon materials, thereby enhancing their intrinsic chemical activities and/or increasing the number of active sites. To further develop high-performance nitrogen-doped $sp^2$-hybridized carbon materials, it is urgently essential to identify active sites of nitrogen-doped $sp^2$-hybridized carbon materials and to understand their interactions with the involved substitute molecules towards specific reactions. Generally, there are four main types of structural nitrogen configurations including pyrrolic nitrogen (N-5), pyridinic nitrogen (N-6), quaternary or graphitic nitrogen (N-Q), and oxidized pyridinic nitrogen atoms (N-X) in nitrogen-doped carbon materials. Apart from N-X, one or more of three other nitrogen centers could be considered to be possible electrochemically active sites[28,36–40]. However, the inevitable nitrogen configuration mixing in carbon materials, greatly limits the precise identification of active nitrogen configurations, the fundamental understanding of their mechanisms for specific reactions, and the further development of high-performance nitrogen-doped carbon materials for energy storage and conversion applications[37,41]. Engineering single nitrogen configuration-doped carbon materials could ideally tackle this problem.

Unfortunately, it seems to be an impossible mission to engineer single nitrogen configuration-doped carbon materials due to the inevitable thermal treatment involved in synthesizing almost all the nitrogen-doped carbon materials via primarily either post-synthetic or in situ-doping methods[21,25]. The post-synthetic doping method mainly involves the thermal treatment of as-synthesized carbon materials with nitrogen-enriched molecules, including ammonia, urea, melamine, and so on[24,42,43]. The inhomogeneous and uncontrolled integration of different nitrogen configurations into carbon materials always occurs in post-synthetic doping processes[39]. In the in situ doping processes achieved mainly by pyrolysis of nitrogen-enriched carbon precursors, such as small molecules, polymers, and biomass[27,40,44–47], thermal treatment above ~600 °C always results in a substantial nitrogen loss and induces uncontrollable conversion among different nitrogen configurations[48,49]; thermal treatment below ~600 °C always yields lowly graphitized nitrogen-doped carbon materials with poor electrical conductivity[50], thereby suppressing their electrochemical activities. Until now, there have been few reports on single or high-purity pyridinic nitrogen-doped[37,51,52] and graphitic nitrogen-doped carbon materials[37] and no reports on single pyrrolic nitrogen-doped carbon materials. Therefore, it is extremely desirable to construct single nitrogen configuration-doped carbon materials, especially pyrrolic nitrogen-doped carbon materials.

Here, we report the preparation of one kind of single nitrogen configuration-doped carbon materials, single pyrrolic N-doped carbon materials (SPNCMs), which are ideal model materials, based on a facile engineering process that combines the low-temperature dehalogenation induced and alkaline-activated pyrolysis of 3-halogenated phenol-3-aminophenol-formaldehyde (X-APF, X = F, Cl, and Br) co-condensed resins; the pyrolyzed dehalogenation-induced coupling reaction occurring between adjacent benzene rings promotes enough graphitization of carbon materials at low temperature, and subsequent alkaline-activated pyrolysis induces the exclusive and complete transformation of $sp^3$-hyrided nitrogen species into $sp^2$-hyrided pyrrolic nitrogen species. The pyrrolic nitrogen content of SPNCMs could be rationally tuned from 0 to 4.22 at.% by alternating the molar ratio of 3-aminophenol to 3-fluorophenol, and the final N-doped carbon materials also exhibit a similar single pyrrolic N-doping state, which was achieved by using 3-chlorophenol or 3-bromophenol as an alternative to 3-fluorophenol. N-doped carbon materials derived from a 3-aminophenol-FPF co-condensed resin with a 3-aminophenol/3-fluorophenol mole ratio of 1/1, namely SPNCMs (F, 1/1), and with up to 4.22 at.% pyrrolic nitrogen, exhibit a high specific capacitance of up to 618 F g$^{-1}$ at 1 A g$^{-1}$, an outstanding rate performance, and an ultrastable cyclic performance, in which the initial specific capacitance was 467 F g$^{-1}$ at 10 A g$^{-1}$, and almost 100% of the capacitance was retained over 30,000 cycles performed with a three-electrode system in 1 M $H_2SO_4$. More importantly, it could be reliably inferred that pyrrolic nitrogen species are highly active pseudo-capacitive sites for nitrogen-doped carbon materials used as electrochemical capacitive materials. The pseudocapacitance of carbon materials is positively dependent on the pyrrolic nitrogen content, as shown by the electrochemical analysis of our single pyrrolic nitrogen-doped carbon materials and nitrogen-absent carbon materials derived from the 3-fluorophenol resin (CMs (FPF)) samples used as ideal model materials, which exhibit both a similar ion diffusion capacity and electrical conductivity. This work presents an ideal model to fundamentally understand the mechanisms of pyrrolic nitrogen species in N-doped carbon materials for energy storage and conversion applications, thus offering a method for designing carbon materials doped with tunable structural nitrogen species for different energy conversion applications.

## Results

**Design principle and synthetic route of SPNCMs.** Figure 1 illustrates the three-step procedure used to synthesize SPNCMs through the combined low-temperature dehalogenation-induced and KOH-activated pyrolysis of X-APF resins, which is described as follows: (1) The synthesis of SPNCMs (F, 1/1), as a typical example, starts with hydrothermal co-condensation of 3-aminophenol, 3-fluorophenol, and formaldehyde at a mole ratio of 1:1:5, and this reaction is catalyzed by ammonia, in which both formaldehyde and ammonia result from hexamethylenetetramine (HMT) decomposition. The synthesis of the resulting crosslinked F-APF (1/1) resin involves the introduction of methylene, a few –CH$_2$–NH– bridging groups and considerable –CH$_2$–N(CH$_2$OH)– bridging groups between 3-aminophenol and 3-fluorophenol monomers, which is achieved by both the ortho- and para-C–H substitution of phenols and N–H substitution of amino groups with methylol groups and the subsequent elimination of $H_2O$ and formaldehyde molecules. (2) The pyrolysis of the F-APF (1/1) resin at 500 °C to form the corresponding nitrogen-doped carbon materials, NCMs (F, 1/1), which are used as the intermediate for SPNCMs, induces the elimination of HF between the C–F bond on one fluorophenol moiety and the C–H bond on one adjacent aminophenol moiety to directly connect two adjacent benzene rings. Meanwhile, all the –CH$_2$–N(CH$_2$OH)– bridging groups between two benzene rings are possibly converted into –CH$_2$–NH– heterocyclic structures by the elimination of formaldehyde or into pyrrolic structures by the

**Fig. 1 Schematic of synthetic procedure of SPNCMs.** The synthetic procedure of SPNCMs based on a strategy of low-temperature dehalogenation-induced and subsequent alkaline-activated pyrolysis of 3-halogenated phenol-3-aminophenol-formaldehyde (X-APF) co-condensed resins.

elimination of dimethyl ester. In the step, partial hydroxyl groups are preserved and some hydroxyl groups are converted to carbonyl groups. (3) The further KOH-assisted pyrolysis at 500 °C enables the transformation of almost all the –CH$_2$–NH– heterocyclic structures into pyrrolic structures by the elimination of –CH$_2$– groups, thus generating the final SPNCMs (F, 1/1) with a large enough graphitization degree. Meanwhile, KOH activation removes some carbon atoms to introduce abundant defects and introduces some hydroxyl and carbonyl groups for final SPNCMs (F, 1/1). When the initial mole ratio of 3-aminophenol/3-fluorophenol is <1, such as 1/3 and 1/7, and even when 3-fluorophenol is replaced with 3-chlorophenol or 3-bromophenol, all the final N-doped carbon materials exhibit a similar single pyrrolic N-doping state.

**Structure and nitrogen species identification of SPNCMs.** Similar to the Raman spectrum of the NCMs (F, 1/1), that of the SPNCMs (F, 1/1) (Fig. 2a) shows a broad D-band attributed to the breathing mode of the six-fold rings at 1353 cm$^{-1}$ and a G-band attributed to the in-plane bond-stretching of C $sp^2$ atoms at 1593 cm$^{-1}$, which are typical feature of $sp^2$-hybridized amorphous carbon materials and are consistent with amorphous carbon in stage 2 transforming from nanocrystalline graphite into $sp^2$ a-C[53,54]. The X-ray diffraction (XRD) patterns of the NCMs (F, 1/1) and SPNCMs (F, 1/1) (Supplementary Fig. 1a) show broadened (002) and (100) diffraction peaks at 20.8–24.4° and 43.5°, respectively[55], further verifying the transition from amorphous F-APF (1/1) resins to amorphous NCMs (F, 1/1) and to amorphous SPNCMs (F, 1/1) with a turbostratic structure. Compared with that of the NCMs (F, 1/1), the pattern of the SPNCMs (F, 1/1) shows that the (002) π-stacking peak shifts to a higher angle and the basal (100) plane is unchanged, indicating the shortened interplanar distance of the turbostratic structure and, in turn, the KOH-enhanced graphitization-like phenomenon for SPNCMs (F, 1/1)[56]. However, the interplanar spacing of the (002) planes of the small nanocrystalline graphite of SPNCMs (F, 1/1) was calculated to be 0.3645 nm, which is far larger than 0.3354 nm, the interplanar spacing of the (002) plane of ideal graphite. The results indicate the long-range disorder and local short-range order structure of the SPNCMs (F, 1/1), whose structure is consistent with the disordered structure comprising randomly stacked $sp^2$ clusters of the relatively small size, according to the scanning electron microscopy (SEM) (Supplementary Fig. 1b), transmission electron microscopy (TEM) (Supplementary Fig. 1c), and high-resolution transmission electron microscopy (HRTEM) (Supplementary Fig. 1d) images of the SPNCMs (F, 1/1). Raman and XRD results mentioned above also exhibit the existence of abundant defects in SPNCMs (F, 1/1). Additionally, energy-

dispersive X-ray spectroscopy (EDS) analysis shows uniform doping of nitrogen species into the turbostratic carbon framework for the SPNCMs (F, 1/1) (Supplementary Fig. 1e–g).

As revealed in Supplementary Fig. 2, the Fourier transform infrared (FTIR) spectra provide some useful information about the structural transformation from the co-condensation resin precursor to the final SPNCMs. The discussion on FTIR spectra is seen in Supplementary Note 1. Furthermore, X-ray photo-electron spectroscopy (XPS) analysis was used to determine the origin of the single pyrrolic nitrogen configuration in the SPNCMs. Compared with the spectrum of the F-APF (1/1) resin, the survey spectra of the NCMs (F, 1/1) and SPNCMs (F, 1/1) preserve similar C1$s$, N1$s$, and O1$s$ peaks, and the F signal almost completely disappears (Fig. 2b), indicating low-temperature defluorination through HF elimination in the first pyrolysis step. The SPNCMs (F, 1/1) preserve the initial N content of the F-APF (1/1) resin (4.27 at.%), with a mere alternation of the nitrogen configuration via the first pyrolysis step and subsequent KOH activation, according to the high-resolution N 1$s$ XPS spectra of the F-APF(1/1) resin (Fig. 2c), NCMs (F, 1/1) (Fig. 2d) and SPNCMs (F, 1/1) (Fig. 2e), and full width at half maximum (FWHM) of corresponding deconvoluted peaks (Supplementary Table 1). The N 1$s$ spectrum of the F-APF (1/1) resin reveals a single peak at 399.25 eV, which corresponds to mixed –CH$_2$–N(CH$_2$OH)– and –CH$_2$–NH– bridging groups; these mixed bridging groups exist because the insufficient formaldehyde molecules that decomposed from hexamethyl tetraamine could not completely substitute for the hydrogens of the primary amine to generate the –CH$_2$–N (CH$_2$OH)– tertiary amine since the number of formaldehyde molecules is lower than that of the available active sites, including the primary amine and *ortho*- and *para*-C–H substitution of phenols. The N 1$s$ XPS spectrum of the NCMs (F, 1/1) intermediate shows the incorporation of a secondary amine (398.9 eV) and N-5 (400.3 eV), which occurs due to the sufficient conversion of the tertiary amine into a secondary amine and N-5 groups by the elimination of formaldehyde and dimethyl ester, respectively. Importantly, the SPNCMs (F, 1/1) merely contain a single pyrrolic nitrogen configuration (400.25 eV). X-ray absorption fine structure (XAFS) analysis of N K edge (Fig. 2f) was conducted to further probe the nitrogen structure of the SPNCMs (F, 1/1). In addition to the 1$s$ → σ* absorption band at 410 eV, there are three well-resolved peaks belonging to 1$s$ → π* features, which range from 400 to 405 eV. The dominant peak at 400.5 eV, namely N-5, is far stronger than the peaks at 402.4 and 404.2 eV, which are assigned to N-Q (valley-N) and N-X, respectively[57], confirming the almost single pyrrolic nitrogen configuration in the SPNCMs (F, 1/1).

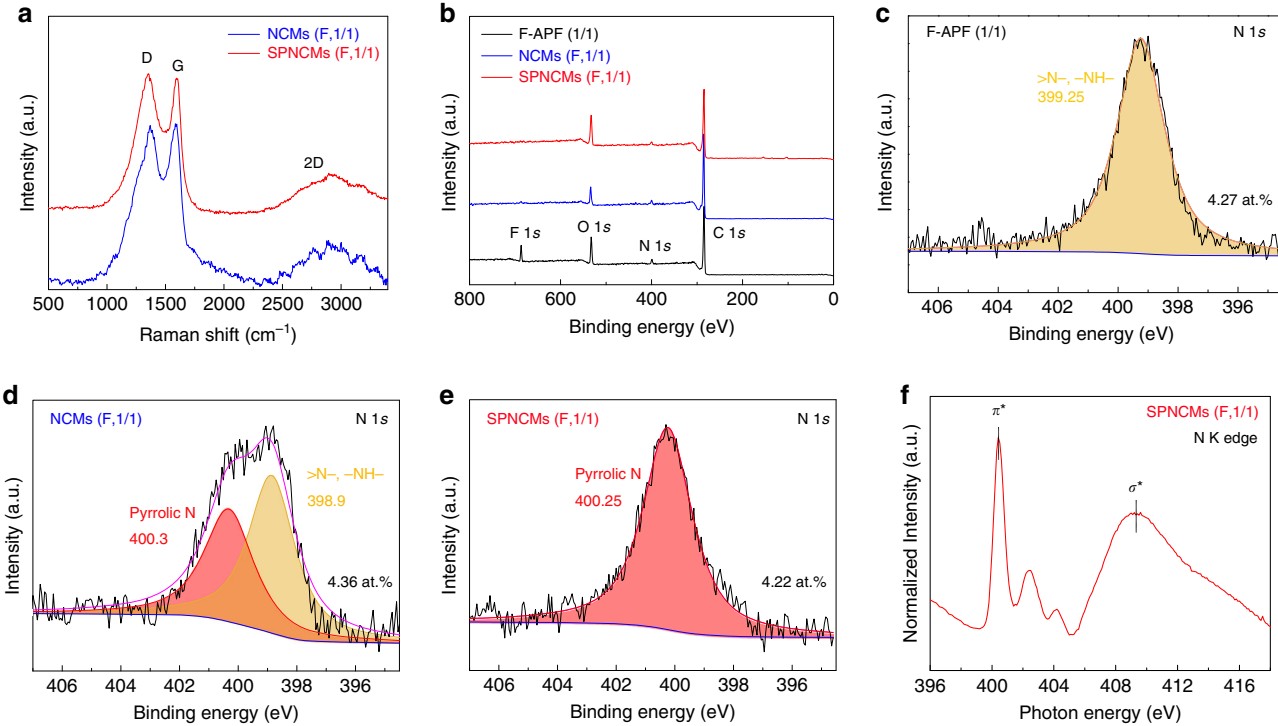

**Fig. 2 Structural characterization of the samples. a** Normalized Raman spectra of the NCMs (F, 1/1) and SPNCMs (F, 1/1). **b** XPS survey spectra of the F-APF resin (1/1), NCMs (F, 1/1), and SPNCMs (F, 1/1). High-resolution N 1s XPS spectra of **c** the F-APF resin (1/1), **d** NCMs (F, 1/1), **e** SPNCMs (F, 1/1). **f** N K edge XAFS spectrum of the SPNCMs (F, 1/1) (the test was performed in the total electron yield (TEY) mode).

Unfortunately, according to XPS analysis, pyridonic nitrogen has a binding energy similar to that of pyrrolic nitrogen, making it difficult to distinguish these two nitrogen configurations. According to previous reports[48,49,58], the two tautomers, 2-hydroxypyridine and α-pyridone, are in equilibrium with one another in pyridonic nitrogen-doped carbon materials. 2-Hydroxypyridine has a chemical environment similar to that of pyridine. Here, 2-hydroxypyridine-type nitrogen has a N 1s binding energy similar to that of pyridine-type nitrogen. Similar with the nitrogen atom of pyrrole, that of α-pyridone provides two p-electrons to the conjugated system, and α-pyridone has a chemical environment similar to that of pyrrole. Therefore, α-pyridone has an N 1s binding energy similar to that of pyrrolic nitrogen. It is reasonable that the coexistence of 2-hydroxypyridine and α-pyridone in pyridonic nitrogen-doped carbon materials means that the N 1s XPS spectrum of pyridonic nitrogen could be deconvoluted into two peaks at ~400.4 and 398.9 eV, which are generally assigned to pyrrolic nitrogen and pyridinic nitrogen. Therefore, we could conclude that the single nitrogen configuration of the binding energy at ~400.3 eV for our nitrogen-doped carbon materials is not pyridone. The previously reported N 1s spectra of nitrogen-doped carbon materials contains mixed nitrogen configurations at ~400.4 and 398.9 eV, which means that the accurate assignment of pyrrolic nitrogen and pyridonic nitrogen is extremely difficult for these nitrogen-doped carbon materials.

Furthermore, to clarify our single nitrogen configuration as pyrrolic nitrogen or pyridonic nitrogen, a 3-fluorophenol-2-hyrdoxypyridine-formaldehyde resin, namely F-HPF (F, 1/1), was designed and synthesized through a procedure similar to that used to prepare the 3-fluorophenol-3-aminophenol-formaldehyde resin, F-APF (1/1), in our manuscript. There are no changes in the pyridine and adjacent hydroxyl groups during polymerization of the phenol-formaldehyde-like resin. In addition, there are no pure pyrrole nor pyridine groups in the F-HPF (1/1) resin.

Therefore, the F-HPF (1/1) resin with only pyridone groups could be used as a model polymer to investigate the pyridone group. Therefore, the FTIR and XPS spectra of the F-HPF (1/1) resin were performed. The FTIR spectrum of the F-HPF (F, 1/1) resin (Supplementary Fig. 3a) shows obvious vibrations of carbonyl group at 1651 cm$^{-1}$ and aromatic C=C group at 1622 cm$^{-1}$, revealing the existence of α-pyridone. The survey spectrum of the F-HPF (F, 1/1) resin (Supplementary Fig. 3b) shows the coexistence of four elements, including carbon, nitrogen, oxygen, and fluorine. 2-hydroxypyridine and α-pyridone could exist in pyridonic nitrogen-doped carbon materials (Supplementary Fig. 3b inset). As expected, the N 1s XPS spectrum of F-HPF (F, 1/1) (Supplementary Fig. 3c, d and Supplementary Table 2) could be deconvoluted into two peaks at 400.35 and 398.95 eV, which are assigned to α-pyridone and 2-hydroxypyridine groups, respectively, and are consistent with the N 1s XPS spectrum of F-HPF (F, 1/1) with the carbonyl group of α-pyridone (531.3 eV) and the hydroxy group of 2-hydroxypyridine (532.95 eV). As mentioned above, we could reliably conclude that the single nitrogen configuration corresponding to the binding energy at ~400.4 eV for our nitrogen-doped carbon materials is not a pyridone group but is actually a pyrrolic nitrogen configuration.

To further determine the dependence of the nitrogen configuration in the final NCMs on the initial mole ratio of 3-aminophenol/3-fluorophenol, the final NCMs with ratios of 1/0, 3/1, 1/3 and 1/7 were also synthesized. When the 3-aminophenol/3-fluorophenol ratio increases to higher than 1, the NCMs (APF) exhibit a single nitrogen configuration, which is a possible pyrrolic nitrogen configuration (400 eV) (Supplementary Fig. 4a and Supplementary Table 3). However, NCMs (APF) used as electrode materials have a poor capacitive behavior, irregular cyclic voltammetry (CV) curves (Supplementary Fig. 4b) and a potential drop of up to 0.5 V in the potential window of 1 V, as shown in the galvanostatic charge/discharge (GCD) curve obtained at a current density of 0.5 A g$^{-1}$ (Supplementary Fig. 4c).

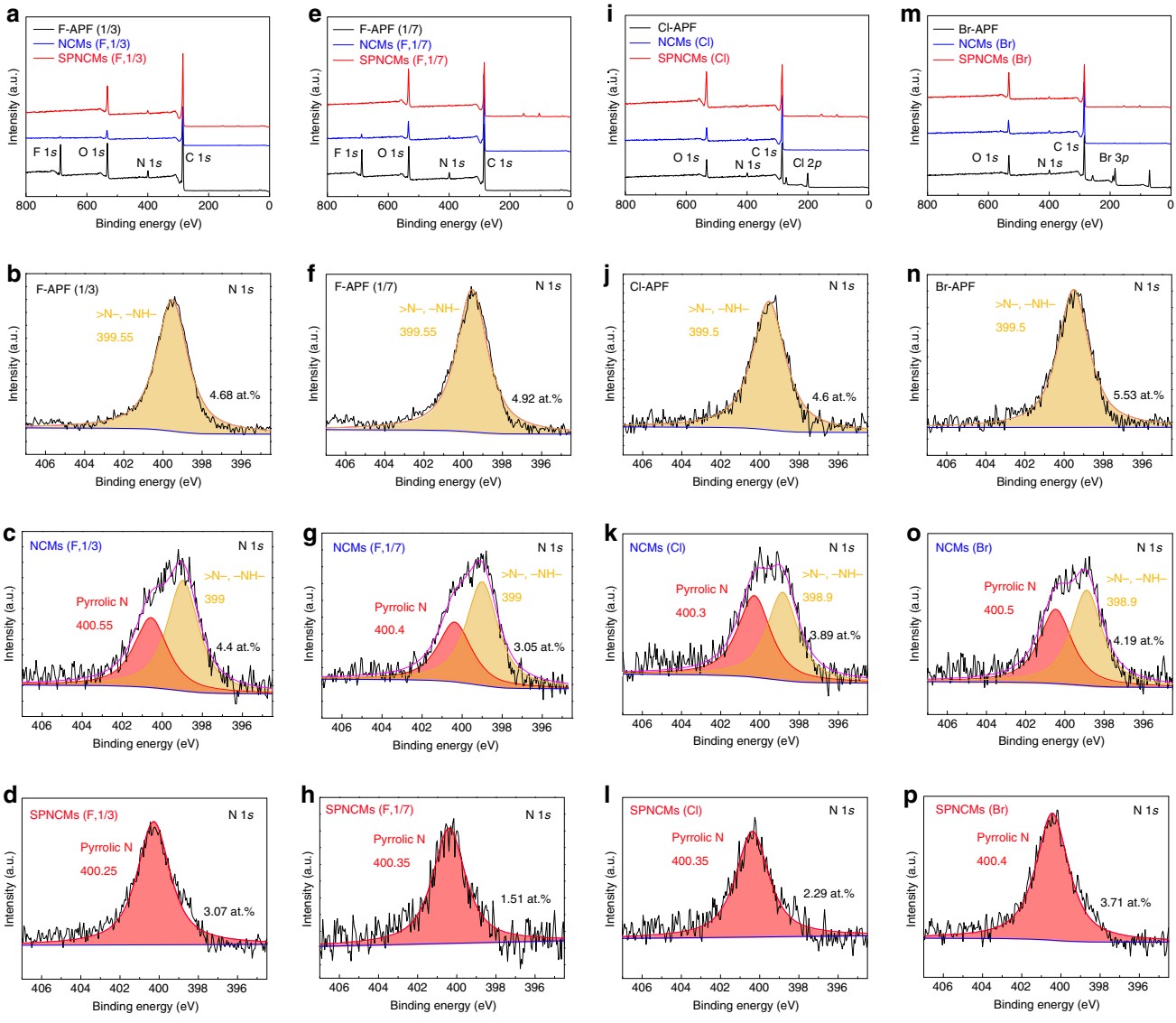

**Fig. 3 XPS spectra of the samples. a**, **e**, **i**, **m** XPS survey spectra of samples. High-resolution N 1*s* XPS spectra for **b** the F-APF resin (1/3), **c** NCMs (F, 1/3), **d** SPNCMs (F, 1/3), **f** F-APF resin (1/7), **g** NCMs (F, 1/7), **h** SPNCMs (F, 1/7), **j** F-APF resin (Cl), **k** NCMs (Cl), **l** SPNCMs (Cl), **n** F-APF resin (Br), **o** NCMs (Br), and **p** SPNCMs (Cl).

These results mean that NCMs (APF) with poor electrical conductivity are not suitable for electrochemical applications, which is possibly attributed to the fact that it is difficult to eliminate hydrogen to couple adjacent benzene rings in the first pyrolysis step. The N 1*s* XPS spectrum of NCMs (F, 3/1) (Supplementary Fig. 4d and Supplementary Table 3) exhibit one peak at 401.7 eV, which possibly corresponds to N-Q. As the ratio decreases from 1/1 to 1/3 and to 1/7, dehalogenation-induced and subsequent KOH-activated pyrolysis ensures the conversion of all the initial nitrogen species into a single pyrrolic nitrogen configuration for the final nitrogen-doped carbon materials, including the SPNCMs (F, 1/1) (Fig. 2e), SPNCMs (F, 1/3) (Fig. 3d), and SPNCMs (F, 1/7) (Fig. 3h) (Supplementary Fig. 5 and Supplementary Table 1) that are attributed to the elimination of sufficient HF molecules between adjacent benzene rings to form pyrrolic rings, and their corresponding N-5 content decreases from 4.22 to 2.96 and to 1.51 at.%. According to the XPS survey and high-resolution spectra (Fig. 3i–p, and Supplementary Table 1), when 3-fluorophenol is replaced with 3-chlorophenol or 3-bromophenol, the final nitrogen-doped carbon materials, including SPNCMs (Cl) and SPNCMs (Br) also exhibit

similar single pyrrolic nitrogen-doping states. Hence, it could be deduced that the two-step pyrolysis of the X-APF co-condensed resin is a reliable and general strategy to engineer SPNCMs.

In addition to a single pyrrolic nitrogen-configuration, oxygen species with high contents were found in all our SPNCMs, which is mainly ascribed to KOH activation. The high-resolution O 1*s* XPS spectra of SPNCMs (F, 1/1), SPNCMs (F, 1/3), and SPNCMs (F, 1/7) are shown in Supplementary Fig. 6a–c and Supplementary Table 4. The O 1*s* XPS spectra of all three of these carbon materials reveal the presence of a hydroxyl/etheric group due to the peak with a binding energy of 533.1–533.3 eV and a carbonyl group due to the peak with a binding energy of 531.9–532.2 eV.

**Ion transfer and electron-conductive capacities of SPNCMs.** It can be generally accepted that N-5 and N-6 are assumed to be related to the Faradaic redox reactions-dependent pseudocapacitance of nitrogen-doped carbon materials[59,60]. However, inevitable nitrogen configuration mixing hampers the ability to distinguish the contribution of N-5 and N-6 pseudocapacitance. Using SPNCMs with different N-5 contents as model carbon materials presents an

opportunity to clarify the pyrrolic nitrogen-dependent pseudocapacitance for nitrogen-doped carbon materials.

According to Fig. 4, the N-5 redox reaction involving the transfer of both a proton and an electron[61], a similar ion transfer capacity and electronic conductivity as well as a single pyrrolic nitrogen configuration are also crucial to precisely elucidate the N-5 pseudocapacitance contribution to carbon materials. Due to an inevitable oxygen residue in carbon materials upon their KOH activation[62], nitrogen-absent CMs (FPF) derived from the combined low-temperature dehalogenation-induced and alkaline-activated pyrolysis of a 3-fluorophenol-formaldehyde (FPF) resin were chosen as the control sample to exclude the contribution of oxygen pseudocapacitance to our SPNCMs. The $N_2$ adsorption/ desorption isotherms of the SPNCMs (F, 1/1), SPNCMs (F, 1/3), SPNCMs (F, 1/7), and CMs (FPF) (Supplementary Fig. 7a) reveal similar type I isotherm features typical of a microporous structure, according to the IUPAC classification, and these four carbon materials have uniform slit micropore geometries at ~0.5–0.7 nm (Supplementary Fig. 7b–d), which means they exhibit similar electrolyte ion transfer capacities. These four carbon materials also show similar excellent electronic conductivity, as inferred from their low equivalent series resistances ~0.74, 0.75, 0.79, and 0.80 Ω for the SPNCMs (F, 1/1), SPNCMs (F, 1/3), SPNCMs (F, 1/7), and CMs (FPF), respectively, which correspond to the $x$-intercepts in the Nyquist plots obtained through electrochemical impedance spectroscopy (EIS) (Supplementary Fig. 8 and Supplementary Fig. 8 inset). All these nearly vertical curves in the low-frequency region indicate excellent capacitive behaviors. As shown in Fig. 2a and Supplementary Fig. 9, the Raman spectra of the SPNCMs (F, 1/1), SPNCMs (F, 1/3), SPNCMs (F, 1/7), and CMs (FPF) have features similar to those of $sp^2$-hybridized amorphous carbon materials, with a broad D-band and G-band.

**Pyrrolic nitrogen content-dependent pseudocapacitance.** Accordingly, these four types of carbon materials could be used as model materials to evaluate the N-5-depenent electrocapacitive performance. All the CV curves obtained at $1\,mV\,s^{-1}$ consist of one nearly rectangular curve, indicating an electrical double layer capacitance (EDLC) behavior and a couple of symmetric peaks at 0.3–0.4 V, derived from the reversible Faradaic redox reactions of N-5 species and/or oxygen (Fig. 5a). Meanwhile, all the

symmetric GCD curves (Fig. 5b) at $1\,A\,g^{-1}$ with obviously distorted linear shapes also indicate typical pseudocapacitive behaviors. As the N-5 content in different carbon materials, ranging from nitrogen-absent CMs (FPF) to SPNCMs (F, 1/7), SPNCMs (F, 1/3), and SPNCMs (F, 1/1) increases, their total specific capacitances increase according to the corresponding CV (Fig. 5a) or GCD (Fig. 5b) curves. All the electrochemical measurements of these carbon materials, ranging from 1 to 20 mV s$^{-1}$ and from 1 to 20 A g$^{-1}$, preserve similar CV (Supplementary Fig. 10a–e) and GCD curves (Supplementary Fig. 11a–e), with a combined EDLC and pseudocapacitance, respectively, and show an outstanding rate performance and slightly decreased capacitances. Particularly, the SPNCMs (F, 1/1) with the highest N-5 content show a specific capacitance of up to 618 F g$^{-1}$ at $1\,A\,g^{-1}$ and maintain a high specific capacitance of 447 F g$^{-1}$, even up to 20 A g$^{-1}$. In addition, the SPNCMs (F, 1/1) both maintain a high specific capacitance of 467 F g$^{-1}$ up to 10 A g$^{-1}$ and reveal an ultrahigh stability over 30,000 cycles (Supplementary Fig. 12a, b). To confirm the electrochemical stability of the SPNCMs (F, 1/1), SEM, TEM, HRTEM, and XPS analyses of the recovered SPNCMs (F,1/1) electrode material were conducted after GCD testing over 30,000 cycles. The SEM (Supplementary Fig. 12c) and TEM (Supplementary Fig. 12d) images demonstrate that the recovered SPNCMs (F,1/1) electrode material maintains the initial morphologic feature of randomly assembled layers. As observed in the HRTEM image (Supplementary Fig. 12e), the recovered SPNCMs (F,1/1) electrode material exhibits a turbostratic carbon structure similar to that of the fresh SPNCMs (F,1/1). EDS analysis (Supplementary Fig. 12f–h) confirms that the recovered SPNCMs (F, 1/1) electrode material maintains the uniform doping of nitrogen species into the turbostratic carbon framework. Furthermore, XPS analysis (Supplementary Fig. 12i, j, and Supplementary Table 5) reveals that the recovered SPNCMs (F,1/1) electrode material maintains a nitrogen content (4.8 at.%) similar to that of the fresh SPNCMs (F,1/1) (4.22 at.%). Unlike the single pyrrolic nitrogen configuration of the fresh SPNCMs (F,1/1) (Fig. 2e), the N 1s XPS spectrum of the recovered SPNCMs (F,1/1) electrode material (Supplementary Fig. 12i) shows the coexistence of pyrrolic nitrogen at 400.4 eV and oxidized pyrrolic nitrogen at 401.9 eV, which could be ascribed to the fact that a redox reaction of pyrrolic nitrogen/electrochemically oxidized pyrrolic nitrogen occurs upon charging–discharging cycles, and the rapid charging–discharging cycles at 10 A g$^{-1}$ of the large current density possibly results in the incomplete use of pyrrolic nitrogen and incomplete reduction of electrochemically oxidized pyrrolic nitrogen. The O 1s XPS spectrum of the recovered SPNCMs (F,1/1) electrode material (Supplementary Fig. 12j) still shows the coexistence of a hydroxyl/etheric group due to the peak with a binding energy of 533.3 eV and a carbonyl group due to

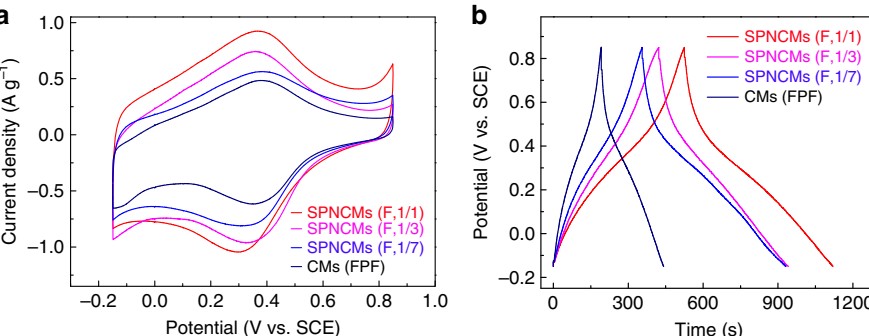

**Fig. 4 Redox reaction.** The redox reaction of pyrrolic nitrogen.

**Fig. 5 Electrochemical evaluation. a** CV test and **b** GCD curves obtained with a three-electrode system at a scan rate of 1 mV s$^{-1}$ and a current density of 1 A g$^{-1}$ in 1.0 M H$_2$SO$_4$ electrolyte for the SPNCMs (F, 1/1), SPNCMs (F, 1/3), SPNCMs (F, 1/7), and CMs (FPF) used as work electrodes.

the peak with a binding energy of 532 eV, which is similar to the O 1*s* XPS spectrum of the fresh SPNCMs (F,1/1) material. The above-mentioned results confirm the high stability of the SPNCMs (F,1/1) electrode material. Generally, N-doped carbon materials exhibit a combined EDLC and faradaic pseudocapacitance and their EDLCs can be quantitatively identified though electrochemical surface area (ESA) tests performed in the non-Faradaic voltage range[63,64]. To further explore the N-5 pseudo-capacitive contribution, based on these nearly rectangular CV curves (Supplementary Fig. 13a–d) ranging from 0 to 0.05 V vs. SCE and the slopes of the scan rate-dependent current density curves (Supplementary Fig. 13e), the EDLCs of the SPNCMs (F, 1/1), SPNCMs (F, 1/3), SPNCMs (F, 1/7), and CMs(FPF) were firstly calculated though ESA tests to be ~125, 263, 211, and 183 $F\,g^{-1}$, respectively, which are highly consistent with their corresponding specific surface areas (SSA) of 450, 950, 873, and 880 $cm^2\,g^{-1}$, respectively. Generally, KOH activation generates nanopores, leading to an increase in defect density for carbon materials. Up to date, it is extremely difficult to precisely control and determine the defect type and density of defects for carbon materials due to the limitations of present characterization. Fortunately, specific surface area represents the density of nanopore-dependent defects for our final carbon materials to a large content. Therefore, the capacitance from these defects could be approximately regarded as the double layer capacitance.

Based on their EDLCs and total specific capacitance of 584.8, 486.7, 412.4, and 301.5 $F\,g^{-1}$ at $1\,mV\,s^{-1}$, the corresponding faradaic pseudocapacitances of the SPNCMs (F, 1/1), SPNCMs (F, 1/3), SPNCMs (F, 1/7), and CMs (FPF) can be calculated as 459.8, 223.7, 201.4, and 118.5 $F\,g^{-1}$, respectively.

Similar to the SPNCMs, the nitrogen-absent carbon material, CMs (FPF), show a typical pseudocapacitive characteristic, with symmetric redox peaks at 0.3–0.4 V vs. SCE, as observed from the CV measurement performed at different scan rates (Supplementary Fig. 10d), demonstrating the reversible Faradaic redox reactions of oxygen species possibly derived from the quinone/hydroquinone redox reaction, which is consistent with a couple of symmetric redox peaks at ~0.4 V vs. SCE for surface-oxidized carbon nanotubes[65] and electrochemically modified glassy carbon[66]. The quinone/hydroquinone redox reaction was described in Fig. 6.

The high-resolution O 1*s* XPS spectrum of the CMs (FPF) (Supplementary Fig. 6d) reveals the presence of a hydroxyl/etheric group due to the peak with a binding energy of 533.3 eV and a carbonyl group due to the peak with a binding energy of 532.1 eV, and these peaks are similar to those observed in the O 1*s* XPS spectra of SPNCMs (F, 1/1), SPNCMs (F, 1/3), and SPNCMs (F, 1/7) (Supplementary Fig. 6a–c). A couple of carbonyl groups could behave as quinones, and a couple of hydroxyl groups behave as hydroquinones. These results mean that these pseudocapacitances of high-oxygen-content SPNCMs (F,1/1, F, 1/3, and F 1/7) with structures similar to those of nitrogen-absent CMs (FPF) could partially derive from the quinone/hydroquinone redox reaction.

So far, due to the great difficulty in precisely determining the structure and number of electrochemically active oxygen species and difficulty in excluding electrochemically inert oxygen species for carbon materials, the merely oxygen-dependent pseudocapacitance of CMs (FPF) exhibiting symmetric redox peaks in their CV measurements performed at different scan rates (Supplementary Fig. 10d) has to be assumed to derive from all the oxygen atoms with identical pseudoca-pacitive contributions. Considering the pseudocapacitance of 118.5 $F\,g^{-1}$ for CMs (FPF) and the oxygen content determined by the XPS analysis (Supplementary Table 6), the specific pseudocapacitance per mole of oxygen atoms can be calculated

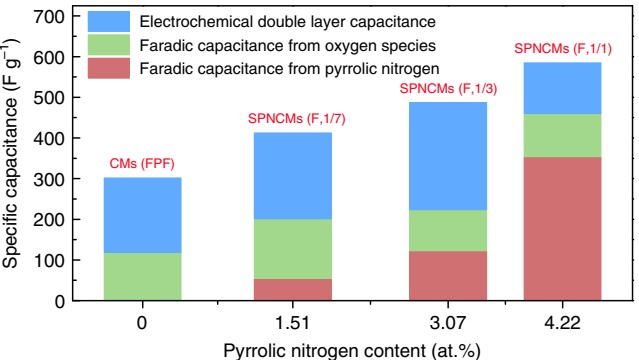

**Fig. 6 Redox reaction.** The quinone/hydroquinone redox reaction.

**Fig. 7 Variations in the capacitances for the samples.** Variations in the pseudocapacitances with respect to pyrrolic nitrogen content in CMs (FPF), SPNCMs (F, 1/7), SPNCMs (F, 1/3), and SPNCMs (F, 1/1).

to be 8471.75 F. Accordingly, the oxygen-dependent specific pseudocapacitances were calculated to be 105.9, 100.8, and 146.3 $F\,g^{-1}$ for the SPNCMs (F, 1/1), SPNCMs (F, 1/3), and SPNCMs (F, 1/7), respectively, and then, their corresponding single pyrrolic nitrogen-dependent pseudocapacitances were calculated to be 353.9, 122.9, 55.1 $F\,g^{-1}$, respectively (Supplementary Table 7, Fig. 7). Therefore, it could be deduced that the pseudocapacitance is positively dependent on the N-5 content in nitrogen-doped carbon materials because of our SPNCMs, which exhibit a similar ion diffusion capacity, electron conductivity and single pyrrolic nitrogen configuration. To further determine whether pyrrolic nitrogen is suitable as an electrochemically active site for Faradic reactions, the SPNCMs (F, 1/1) after charging and discharging at a current density of 1 $A\,g^{-1}$ in a 1.0 M $H_2SO_4$ solution were recovered to investigate the electrochemical redox change of the SPNCMs (F, 1/1). The nitrogen content of the fresh SPNCMs (F, 1/1), the recovered SPNCMs (F, 1/1) after charging, and the recovered SPNCMs (F, 1/1) after discharging is 4.22, 3.97 and 3.69 at.%, respectively, based on XPS analysis, which means that there is no obvious change in the nitrogen content for SPNCMs (F, 1/1) before and after performing GCD tests. The high-resolution N 1*s* spectra of these three samples (Fig. 2e, Supplementary Table 1, Supplementary Fig. 14a, 14b, and Supplementary Table 8) show the dramatic conversion of pyrrolic nitrogen (~400.2 eV) into oxidized pyrrolic nitrogen (~401.9 eV) after charging and the corresponding obvious reverse conversion process after discharging, which confirms that pyrrolic nitrogen is an electrochemically active site for nitrogen-doped carbon materials exhibiting pseudocapacitance. Meanwhile, the oxygen content of the fresh SPNCMs (F, 1/1), the recovered SPNCMs (F, 1/1) after charging, and the recovered SPNCMs (F, 1/1) after discharging is 16.69, 18.64, and 15.45 at.%, respectively, based on XPS analysis, which means that there is no obvious change in the oxygen content for SPNCMs (F, 1/1) before and after performing GCD tests. The high-resolution O 1*s* spectra of these three samples (Supplementary Fig. 6a, Supplementary Table 4, Supplementary Fig. 14c, 14d, and Supplementary Table 8) show the conversion of a hydroxyl group (~533.3 eV) into a carbonyl group (~532.1 eV) after charging and the corresponding partial reverse conversion after discharging, which is possibly ascribed to quinone/hydroquinone redox reaction.

## Discussion

In summary, we report SPNCMs with tunable nitrogen contents used as model nitrogen-doped carbon materials and prepared though two-step pyrolysis of X-APF co-condensed resins. Moreover, it could be inferred that N-5 species are highly active pseudocapacitive sites for nitrogen-doped carbon materials, which was inferred based on the result that the pseudocapacitance of the carbon materials is positively dependent on the N-5 content, as shown by the electrochemical analyses of the SPNCMs and nitrogen-absent CMs (FPF). The discovery of SPNCMs as ideal model carbon materials not only identifies N-5 species as active sites and their mechanisms toward energy storage and conversion applications, but also offers a method for designing carbon materials doped with tunable structural nitrogen species for different energy conversion applications.

## Methods

**Materials**. 3-Aminophenol (3-AP; >98.5%), HMT (99.5%), and 3-Bromophenol (98%) were purchased from TCI, Alfa-Aesar, and Aladdin Industrial Corporation, respectively. 3-Fluorophenol (98%), 3-Chlorophenol (98%), and potassium hydroxide (KOH; GR; 95%) were purchased from Shanghai Macklin Biochemical Co., Ltd. Sulfuric acid and hydrochloric acid were supplied by Tianjin Kermel Chemical Reagent Co., and polyvinylidene difluoride (PVDF) was obtained from Guangdong Candlelight Amperex Technology Ltd.

**Materials preparation**. Supplementary Tables 9–11 show the details of the synthesis parameters of the 3-halogenated phenol-3-aminophenol-formaldehyde (X-APF) co-condensed resin (X = F, Cl, and Br), the corresponding pyrolyzed intermediates and the final nitrogen-doped carbon materials after activation.

Typically, certain amounts of 3-aminophenol, 3-fluorinephenol (3-chlorophenol or 3-bromophenol), and HMT were mixed in 80 ml deionized water, stirred for 1 h at room temperature, and then transferred into a Teflon-lined stainless-steel autoclave, followed by heating at 160 °C for 4 h. The brown product was purified with distilled water several times through a suction device and then dried at 60 °C for 12 h.

Subsequently, the first pyrolysis step was applied to the corresponding cocondensed resins were conducted. The as-obtained materials were placed into a tube furnace, heated to 500 °C with a ramp rate of 1 °C min⁻¹ and then held for 4 h under pure nitrogen gas (99.999%) at a flow rate of 200 mL/min.

Finally, the mixture of the above-mentioned pyrolyzed product and KOH was heated at 500 °C for 8 h until the whole mixture turned into a paste, followed by washing with an HCl solution and deionized water. After being dried, the final product after activation was obtained.

The 3-Fluorophenol-2-hyrdoxypyridine-formaldehyde resin, namely, F-HPF (F, 1/1), was designed and synthesized through the following procedure: 0.085 g of 2-hydroxypyridine, 0.1 g of 3-fluorinephenol, and 0.074 g of HMT were mixed in 80 ml deionized water, stirred for 1 h at room temperature, and then transferred into a Teflon-lined stainless-steel autoclave, followed by heating at 180 °C for 24 h. The product was purified with distilled water several times through a suction device and then dried at 60 °C for 12 h.

**Characterization**. The structures of the samples were characterized on a HITACHI HT 7700 transmission electron microscope, with an accelerating voltage of 100 kV. HRTEM and EDS analyses were performed on a JEM-2010 instrument operated at an accelerating voltage of 200 kV. SEM images were acquired on a SUPRA 55 field-emission scanning electron microscope with an accelerating voltage of 200 kV. The powder XRD data was conducted on a RIGAKU X-ray diffractometer with CuKa radiation (l = 0.15418 nm) at 40 kV and calibrated using the RIGAKU Silicon-640 as standard sample before the measurement. Raman spectra were recorded on a Renishaw in Via Raman microscope with an Ar ion laser at the excitation wavelength of 532 nm. XPS analyses were performed with an ESCALAB 250 Xi spectrometer by employing an Al Kα X-ray source. The pass energies for XPS survey spectra and high-resolution XPS spectra test of all the samples are 100.0 and 30.0 eV, respectively. All the high-resolution XPS spectra were fitted with the Lorentz/Gauss mixing ratios fixed at 0.8, sample charging was corrected by using the C 1s binding energy at 284.8 eV, and Shirley-type background subtraction was applied. As listed in Supplementary Tables 1–5, 8, the FWHMs of N 1s were determined to be 1.9–2 eV, and the FWHMs of O 1s are determined to be ~1.8 eV. FTIR spectra were recorded on a Nicolet IS 10 infrared spectrometer. N K edge XAFS spectrum of the SPNCMs (F, 1/1) was tested in the total electron yield (TEY) mode by BSRF 4B7B.

**Electrochemical measurement**. The working electrodes were prepared by pressing the mixture of samples, acetylene carbon black, and PVDF binder at the mass ratio of 85/10/5 in 1 M $H_2SO_4$. The above mixture ~1 mg cm⁻²) was coated onto a 1 × 1 cm platinum net and dried at 120 °C for 12 h in vacuum.

Electrochemical performance of supercapacitor was firstly tested in a three-electrode system in 1 M $H_2SO_4$. The platinum foil and $Hg/Hg_2Cl_2$ (SCE) in 1 M $H_2SO_4$ were used as counter electrode and reference electrode, respectively. EIS spectra were recorded at the frequency range of 0.01–10⁵ Hz. CV curves were measured at the scanning rate of 1–20 mV s⁻¹. Both EIS and CV tests were carried out on a CHI 660E electrochemical workstation. The constant current charge–discharge tests were performed with a computer-controlled cycling equipment (Land CT2001A, China) in the potential rage of −0.15 to 0.85 V vs. SCE in 1 M $H_2SO_4$.

**Calculation equations for supercapacitor**. The capacitance calculation of three electrode system according to Eq. (1) with the formula:

$$C = \frac{S_{Area}}{2\nu\Delta V} \tag{1}$$

where $S_{Area} = \oint IdV$ is the loop area from the CV test; $\nu$ is the scan rate; and $\Delta V$ on behalf of the potential window within one cycle.

The specific capacitance was estimated from the galvanostatic charge–discharge curves, by Eq. (2):

$$C_S = \frac{I \times \Delta t}{m \times \Delta V} \tag{2}$$

where $I$, $\Delta t$, $m$, and $\Delta V$ stand for the discharge current, discharge time, the total mass of active material on three electrodes.

The EDLC capacitance was determined by electrochemical surface areas (ESA) in the non-Faradic voltage range, by Eqs. (3) and (4):

$$ESA = \frac{I_1 - I_2}{\nu_1 - \nu_2} \tag{3}$$

$$C_{EDLC} = \frac{ESA}{M} \tag{4}$$

where $I$ is the current density; $\nu$ is the scan rate; and $M$ is the mass of active material per unit area.

## Data availability

The data that support the findings of this study are available from the corresponding author upon reasonable request. Source data are provided with this paper.

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

## Acknowledgements

This work is financially supported by the National Natural Science Foundation of China (51503178, 51631001, 51590882), National Key R&D Program of China (2017YFA0206301, 2016 YFA0200102) and Natural Science Foundation of Beijing Municipality (219001). We also thank Professor Ping Xu from Harbin Institute of Technology, Professor Guangjie Shao, Professor Zhisheng Zhao, Mr. Hongchao Wang from Yanshan University for their help with the electrochemical, XRD and BET analysis.

## Author contributions

K.T., W.G., Y.H., and H.W. conceived the concept, designed the experiments, analyzed the experimental data, discussed the related results, and wrote the manuscript. K.T., J.W., L.C., and W.Y. synthesized the resins and the corresponding nitrogen-doped carbon materials and performed electrochemical measurements. S.L. analyzed N K edge XAFS spectrum of the samples. W.L., F.W., X.L., and Z.X. performed Raman, TEM, and XRD analysis. Z.W. performed some electrochemical analysis.

## Competing interests

The authors declare no competing interests.
