## [Peer Review File · Nature Communications]

Reviewers' Comments:

Reviewer #1:

Remarks to the Author:

In this work, Tian et al. reported their efforts on the design and synthesis of single pyrrolic N-doped carbon materials (SPNCMs). Their results revealed that the SPNCMs (F, 1/1) with up to 4.22 at.% pyrrolic nitrogen had the high specific capacitance performances in 1 M H₂SO₄. This is an interesting topic and shows some interesting results. However, some critical issues are missing. Therefore, I would like to recommend it for publication after the revision as follows.

1. In the Introduction section, the authors claimed that "Apart from N-X, one or more of three other nitrogen centers could be considered as possible electrochemically active sites.". To help enhance the readability of the paper, the proper references are needed.
2. How much is the ratio of intensity of D/G bands before nitrogen doping?
3. In page 5, the authors claimed that the XRD results verified the transition from amorphous F-APF (1/1) resins to amorphous NCMs (F, 1/1) to amorphous SPNCMs (F, 1/1). However, in page 6, the authors stated that "About 0.34-nm interplanar spacing corresponding to the (002) plane of graphitic layers confirms the disordered structure randomly stacked by small domains of microcrystalline graphite according to SEM (Supplementary Fig. 1b), TEM (Supplementary Fig. 1c) and HRTEM (Supplementary Fig. 1d) images of SPNCMs (F, 1/1)". These two conclusions seem to be in contrast. The authors should check this. In addition, how did the authors observe the domains of microcrystalline graphite through the SEM and TEM images at such low resolution level?
4. The authors aimed to understand the contribution of nitrogen species in N-doped carbon materials and they claimed that pyrrolic nitrogen species are exactly highly active pseudocapacitive sites for their materials. However, the O species were detected in all samples, which are confirmed by XPS data (Figure 3). Especially, the SPNCMs exhibited much larger oxygen content than its precursor (NCMs, Supplementary Table 2). Since there have been plenty of previous reports that show O-doped carbons as active materials. Therefore, the effects of oxygen species on the performance must be considered, and the O 1s XPS data should be provided.
5. The EIS data are suggested to be fitted by using an equivalent circuit model.
6. The authors stated that the SPNCMs (F, 1/1) had ultrahigh stability over 30,000 cycles during the tests. To confirm this, the SEM, TEM, HRTEM and XPS characterizations on the samples obtained after the stability tests are required.
7. Some important references on the latest advances of carbon materials (for example, Chem. Soc. Rev., 2014, 43, 2572; Acc. Chem. Res. 2014, 47, 1186–1198; Acc. Chem. Res. 2017, 50, 2470–2478; Nat. Commun. 2018, 9, 1460; J. Am. Chem. Soc. 2019, 141, 10677; Nat. Commun. 2018, 9, 5309; Nat. Commun. 2018, 9, 3376; Angew. Chem. Int. Ed. 2018, 57, 774–778; and Chem. Rev. 2018, 118, 7744–7803;) are suggested to be cited in the manuscript.

Reviewer #2:

Remarks to the Author:

The authors reported preparation method of nitrogen doped carbon materials with single XPS peak which was assigned as a pyrrolic nitrogen. They also reported that the amount of XPS peak can be concluded to correspond to the pseudocapacitance. If the assignment is reliable, the work may have a significance in terms of tunable controllability of specific nitrogen species in carbon materials and clarification of the pseudocapacitive active site. However, the conclusion strongly relies on the XPS assignment and the evidence of pyrrolic nitrogen species is currently weak, which does not convince the general reader of Nature Communications. Moreover, the redox peak assignment is ambiguous. Authors did not clearly mention about the "capacitance from oxygen species". Also, the well-known quinone redox is not discussed. The component contribution shown in Fig. 5 should be carefully made

by the analysis of the sample before and after the electrochemical measurements. Thus, I recommend the manuscript to be published if the assignment becomes reliable after the major revision. Specific points the authors should consider are listed below:

(1) Assignment of pyrrolic nitrogen

Authors assigned their observed N1s peak at 400.2-400.25 eV as pyrrolic nitrogen. This is well known species and thus normally the assignment is not so problematic. However, in this manuscript, this peak component is a key point of the main conclusion and thus authors should carefully assign it with enough evidence because the other type of nitrogen is known to also show the peak at similar energy position (and the possibility of these other nitrogen species cannot be excluded by the character of precursor because authors conducted the sample treatment for the precursor e.g. heating). Specifically, the peak at 400.2 eV is reported to appear as core level peak of pyridonic nitrogen (Carbon 33, 1641–1653 (1995), Energy Fuels, 8, 896–906 (1994), Fuel, 78, 923–932 (1999), ACS Nano 8, 6856–6862 (2014).). Also, the peak at 400.7 eV is reported as hydrogenated nitrogen species such as hydrogenated pyridinic functionalities (J. Phys. Chem. C 120, 29225–29232 (2016).). Therefore, the additional evidence to classify these species should be shown. For example, if the authors show IR spectrum with no correlation of –OH amount and N1s peak intensity, authors can safely avoid the mixing of pyridonic nitrogen. Also, if authors show the drastic decrease of N1s peak by heating at higher temperature, authors may discuss that this is not come from hydrogenated pyridinic nitrogen (because pyrrolic nitrogen is known to be unstable against heating). These are “at least” required evidences, further strong and direct evidences (e.g. STM, TEM, or STEM measurements or other spectroscopic evidences with theoretically calculated models) are required to conclude the main issue of the manuscript and convince the reader of Nature Communications.

(2) Redox peak

Authors described about 0.3-0.4 V redox peak as “reversible Faradaic redox reaction of N-5 species and/or oxygen”. It is not clear about Faradaic redox reaction of oxygen because authors did not show any possible exact reaction with evidence (e.g. at least reference should be shown). It seems quinone/hydroquinone redox reaction. In any case, authors should discuss the redox reaction at 0.3-0.4 V more carefully because this is a part of main message of this manuscript.

(3) Fig. 5

Authors construct Fig. 5 partly based on oxygen content of XPS peak. However, the sample character changes significantly after immersion the sample in electrochemical cell and CV measurements. Thus authors should carefully check the sample condition before and after the measurement otherwise at worse case it causes misleading for general readers of Nature Communications.

(4) Minor point

Authors should show the detail of XPS measurement and analysis conditions more clearly. Specifically, for example, pass energy used for the measurement, peak fitting process (how to determine the width of each peak and what is the peak width), and about charge-up correction (if any).

Reviewer #3:

Remarks to the Author:

This study reported single pyrrolic N-doped carbon materials (SPNCMs) with a tunable nitrogen content from 0 to 4.22 at.% based on a combined strategy, and inferred that pyrrolic nitrogen species are exactly highly active pseudocapacitive sites for nitrogen-doped carbon materials. The authors try to set an ideal model to understand the contribution of nitrogen species in N-doped carbon materials, offering the principle of materials design for different energy conversions. However, the investigation on the experimental part is very weak (See Below). The quality of this work cannot meet the required

standard of Nature commutation. Therefore, the reviewer cannot recommend its acceptance for publication.

Specific comments

1. The chemical structure of NCMs and SPNCMs should be further demonstrated by more physical characterization except XPS.
2. As KOH activation was involved in the synthetic procedure, may defect structure and effects should also be considered for the electrochemical performance. It is hard to say the materials developed in this work as ideal model catalysts, particularly after the pyrolysis.
3. Some of the XPS spectra are not well fitted (eg., Figure 2c and Figure 3k). The basic rules of the deconvolution should be obeyed. At least, the half width of the each peak for one element should be consistent!
4. Page 2, Line 52. "The authors stated that "it seems to be an impossible mission to engineer single nitrogen configuration-doped carbon materials due to an inevitable thermal treatment involved in synthesizing almost all the nitrogen-doped carbon materials primarily via either post-synthetic or in situ doping methods 5,9,23." However, Ref. 21 has already reported the synthesis of pure pyridinic N or graphitic N doped HOPG. Also, the Ref. 23 reported on nanoporous carbon spheres, rather than doping type of N.
5. Some of the previous literature reported that pyridinic (pyrrolic also) N species can be transferred into the graphitic N during the pyrolysis, and that the pyrrolic N is not stable during the electrochemical catalytic process, how about the stability of the catalyst developed in this work?
6. The authors stated that they reported one kind of single pyrrolic N-doped carbon materials (SPNCMs) as ideal model materials for supercapacitor. However, the XPS results show a high content of oxygen, and Raman spectra show the presence of numerous defects. Both of the oxygen functional groups and defects will affect the (pseudo-) capacitance, which should be distinguished from the contribution of the pyrrolic N.
7. The graphitization degree should also be determined (e.g. by XRD), which may show a relatively low graphitization degree with respect to the HOPG doped by pyrrolic N only previously-reported by Nakamura et al. (i.e., Ref. 21). The so-called single pyrrolic N-doped carbon (SPNCM) is inferior to its HOPG counterparts.
8. In order to claim the supercapacitance from the single pyrrolic N-doped carbon materials (SPNCMs), the single pyridinic N-doped, and single graphitic N-doped SPNCM counterparts need to be prepared as references so that the oxygen and defect effects can be neglected.
9. Some Refs cited in this study is not suitable, for example:
Page 2, Line 33. "...oxygen reduction reaction (ORR)6-9, CO2 reduction reaction10-12, oxygen evolution reaction (OER)13-15, hydrogen evolution reaction (HER)16, nitrogen reduction reaction17-20, and so on...", however, the Ref. 8 focused on "Fe-Nx" active sites for ORR, while Ref. 17 studied on the catalysis of CO2 reduction.

Reviewers' comments:

Reviewer #1 (Remarks to the Author):

In this work, Tian et al. reported their efforts on the design and synthesis of single pyrrolic N-doped carbon materials (SPNCMs). Their results revealed that the SPNCMs (F, 1/1) with up to 4.22 at.% pyrrolic nitrogen had the high specific capacitance performances in 1 M H₂SO₄. This is an interesting topic and shows some interesting results. However, some critical issues are missing. Therefore, I would like to recommend it for publication after the revision as follows.

1. In the Introduction section, the authors claimed that “Apart from N-X, one or more of three other nitrogen centers could be considered as possible electrochemically active sites.”. To help enhance the readability of the paper, the proper references are needed.

Response:

Thanks for the valuable reminder from the referee. The proper references on pyrrolic, pyridinic and graphitic nitrogen as possible electrochemically active sites have been added (Adv. Energy Mater. 7, 1701456 (2017); Angew. Chem. Int. Ed. 56, 7847-7852 (2017); Science 351, 361-365 (2016); ACS Nano 8, 6856-6862 (2014); Energy Environ. Sci. 5, 7936-7942 (2012); Adv. Funct. Mater. 28, 1800499 (2018)).

2. How much is the ratio of intensity of D/G bands before nitrogen doping?

Response:

Our single pyrrolic nitrogen carbon materials are derived from carbonization and subsequent activation of nitrogen-enriched polymer precursor, aminophenol formaldehyde resin, therefore the sample before nitrogen doping is NCMs (F, 1/1) for the final single pyrrolic nitrogen-doped carbon materials, SPNCMs (F, 1/1). The Raman spectrum of NCMs (F, 1/1) has been added in Fig. 2a, and the corresponding ratio of intensity of D/G bands was determined to be 0.99, slightly lower than that of SPNCMs (F, 1/1) (1.03), possibly because alkaline activation induces more defective sites when nitrogen-containing functionals are completely converted to single pyrrolic nitrogen configuration.

3. In page 5, the authors claimed that the XRD results verified the transition from amorphous F-APF (1/1) resins to amorphous NCMs (F, 1/1) to amorphous SPNCMs (F, 1/1). However, in page 6, the authors stated that “About 0.34-nm interplanar spacing corresponding to the (002) plane of graphitic layers confirms the disordered structure randomly stacked by small domains of microcrystalline graphite according to SEM (Supplementary Fig. 1b), TEM (Supplementary Fig. 1c) and HRTEM (Supplementary Fig. 1d) images of SPNCMs (F, 1/1).”. These two conclusions seem to be in contrast. The authors should check this. In addition, how did the authors observe the domains of microcrystalline graphite through the SEM and TEM images at such low resolution level?

Response:

Thanks for the valuable reminder from the referee. Indeed, the domains of microcrystalline graphite could be observed through HRTEM image and not be observed through the SEM and TEM images at a low-resolution level. We have noticed now this misuse and revised this

description as that small domains of microcrystalline graphite were observed through HRTEM (Supplementary Fig. 1d) image of SPNCMs (F, 1/1). Meanwhile, we use a clearer HRTEM (Supplementary Fig. 1d) image of SPNCMs (F, 1/1) to replace the previous edition. However, it is still difficult to accurately determine interplanar spacing of the (002) plane of small microcrystalline graphite. We have noticed about 0.34-nm interplanar spacing with relatively large error. Therefore, interplanar spacing of the (002) plane of small microcrystalline graphite of SPNCMs (F, 1/1) was calculated to be 0.3645 nm, far larger than 0.3354 nm, interplanar spacing of the (002) plane of ideal graphite. Combined with the broadening 100 and 002 peaks of SPNCMs (F, 1/1) (Supplementary Fig. 1a), the results mentioned above verify SPNCMs (F, 1/1) with turbostratic carbon, consistent with our HRTEM result and previous report of carbon materials with turbostratic structure derived from phenol-formaldehyde resin (Mechanical properties and chemistry of carbonization of Phenol formaldehyde resin, Carbon Volume 24, Issue 5, 1986, Pages 575-580). Thanks for the valuable reminder from the referee, “amorphous SPNCMs (F, 1/1)” has been revised as “SPNCMs (F, 1/1) with turbostratic structure”.

4. The authors aimed to understand the contribution of nitrogen species in N-doped carbon materials and they claimed that pyrrolic nitrogen species are exactly highly active pseudocapacitive sites for their materials. However, the O species were detected in all samples, which are confirmed by XPS data (Figure 3). Especially, the SPNCMs exhibited much larger oxygen content than its precursor (NCMs, Supplementary Table 2). Since there have been plenty of previous reports that show O-doped carbons as active materials. Therefore, the effects of oxygen species on the performance must be considered, and the O 1s XPS data should be provided.

Response:

Thanks for the valuable reminder from the referee. CV curve of nitrogen-absent CMs (FPF) shows a couple of symmetric redox peaks at 0.3-0.4 V vs SCE, demonstrating reversible Faradaic redox reactions of oxygen species possibly derived from quinone/hydroquinone redox reaction, which is consistent with a couple of symmetric redox peaks at approximately 0.4 V vs SCE for surface-oxidized carbon nanotube (*Journal of The Electrochemical Society*, 2006,153(10), A1823-A1828) and Electrochemically Modified Glassy Carbon (*Journal of The Electrochemical Society*, 2000, 147 (7), 2636-2643). Quinone/hydroquinone redox reaction was described in equation 2:

This result means that high-oxygen-content SPNCMs with similar structures to nitrogen-absent CMs (FPF) could also contribute quinone/hydroquinone redox reaction to pseudocapacitance. Herein, O 1s XPS spectra of our carbon materials including SPNCMs (F, 1/1), SPNCMs (F, 1/3), SPNCMs (F, 1/7) and CMs (FPF) in Supplementary Figure 6 have been added into Supplementary Information. O 1s XPS spectra of all these four carbon materials reveals the presence of hydroxyl/etheric group of the binding energy at 533.1-533.3eV and carbonyl group of the binding energy at 531.9-532.2eV. A couple of carbonyl groups could behave as quinone, and a couple of hydroxyl groups behave as hydroquinone. These XPS results, combined with a couple of symmetric redox peaks in CV curve of nitrogen-absent CMs (FPF), confirm that the

pseudocapacitance of our single pyrrolic nitrogen-doped carbon materials could partially stem from possible oxygen-dependent quinone/hydroquinone redox reaction.

Supplementary Figure 6. High-resolution O 1s XPS spectra of SPNCMs (F, 1/1) (a), SPNCMs (F, 1/3) (b), SPNCMs (F, 1/7) (c), and CMs (FPF) (d).

5. The EIS data are suggested to be fitted by using an equivalent circuit model.

Response:

Thanks for the advice from the referee. The EIS data of SPNCMs (F, 1/1) has been fitted by using an equivalent circuit model, and a simplified equivalent circuit model and the simulated Nyquist plot have been added into Supplementary Figure 8. (b).

With respect to this equivalent circuit model, the solution resistance (R_s) and the charge-transfer resistance (R_{ct}) was determined to be 0.8 ohm and 2.1 ohm, respectively. In addition, C_{dl} and Z_w represent the double-layer capacitance and the Warburg impedance.

Supplementary Figure 8. (a) Complex-plane plots of electrochemical impedance spectroscopy (EIS) for SPNCMs (F, 1/1), SPNCMs (F, 1/3), SPNCMs (F, 1/7), and CMs (FPF) samples in frequency ranged from 0.01 to 100000 Hz in 1.0 M H₂SO₄. The inset shows the corresponding details at high-frequency ranges. (b) Nyquist plots (red dots) and Nyquist plot simulation (green boxes) of EIS for SPNCMs (F, 1/1); the inset displays the fitted equivalent circuit of the experimental data.

6. The authors stated that the SPNCMs (F, 1/1) had ultrahigh stability over 30,000 cycles during the tests. To confirm this, the SEM, TEM, HRTEM and XPS characterizations on the samples obtained after the stability tests are required.

Response:

As shown in Supplementary Figure 13a, SPNCMs (F, 1/1) preserves high specific capacitance of 467 F g⁻¹ up to 10 A g⁻¹ and reveals ultrahigh stability over 30,000 cycles during the GCD tests at the large current density of 10 A g⁻¹. In order to confirm the electrochemical stability of SPNCMs (F, 1/1), SEM, TEM, HRTEM, and XPS analysis of the recovered SPNCMs (F,1/1) electrode material after GCD test over 30,000 cycles were performed. SEM and TEM images demonstrate that the recovered SPNCMs (F,1/1) electrode material preserves the initial morphologic feature of randomly assembled layers. As observed in HRTEM image, the recovered SPNCMs (F,1/1) electrode material preserves HRTEM image exhibits similar turbostratic carbon structure to fresh SPNCMs (F,1/1) material. EDS analysis confirms the recovered SPNCMs (F, 1/1) electrode material maintains uniform doping of nitrogen species into the turbostratic carbon framework (Supplementary Figure 13). Furthermore, XPS analysis reveals that the recovered SPNCMs (F,1/1) electrode material preserves similar nitrogen content (4.8 at.%) to the fresh SPNCMs (F,1/1) material (4.22 at.%). Different from the single pyrrolic nitrogen configuration of the fresh SPNCMs (F,1/1) material (Fig. 2e), the N 1s XPS spectrum of the recovered SPNCMs (F,1/1) electrode material shows the coexistence of pyrrolic nitrogen at 400.4 eV and oxidized pyrrolic nitrogen at 401.9 eV, which could be ascribed to that there is the redox reaction of pyrrolic nitrogen/electrochemically oxidized pyrrolic nitrogen upon charging-discharging cycles, and rapid charging-discharging cycles at 10 A g⁻¹ of the large current density possibly results in incomplete use of pyrrolic nitrogen and incomplete reduction of electrochemically oxidized pyrrolic nitrogen. The O 1s XPS spectrum of the recovered SPNCMs (F,1/1) electrode material still shows the coexistence of hydroxyl/etheric group of the binding energy at 533.3 eV and carbonyl group of the binding energy at 532 eV, similar to O 1s XPS spectrum of the fresh SPNCMs (F,1/1) material. The results mentioned above confirm high stability of the SPNCMs (F,1/1) electrode material.

Supplementary Figure 13. Gravimetric capacitance (a) and capacity retention (%) (b) of the SPNCMs (F, 1/1) electrodes in 1.0 M H₂SO₄ solution in a three-electrode system at a current density of 10 A g⁻¹ over 30,000 cycles, SEM (a), TEM (b), HRTEM (c), HAADF-STEM (d) images, the elemental carbon (d) and nitrogen (f) mapping images, high-resolution XPS spectra for N 1s (c) and O 1s (d) of the recovered SPNCMs (F, 1/1) electrodes after aforementioned GCD test above over 30,000 cycles.

7. Some important references on the latest advances of carbon materials (for example, Chem. Soc. Rev., 2014, 43, 2572; Acc. Chem. Res. 2014, 47, 1186–1198; Acc. Chem. Res. 2017, 50, 2470–2478; Nat. Commun. 2018, 9, 1460; J. Am. Chem. Soc. 2019, 141, 10677; Nat. Commun. 2018, 9, 5309; Nat. Commun. 2018, 9, 3376; Angew. Chem. Int. Ed. 2018, 57, 774–778; and Chem. Rev. 2018, 118, 7744–7803;) are suggested to be cited in the manuscript.

Response:

Thanks for the referee's valuable advice. We have retrieved and read detailly the latest advances of carbon materials, and related references have been cited in our revised manuscript.

Reviewer #2 (Remarks to the Author):

The authors reported preparation method of nitrogen doped carbon materials with single XPS peak which was assigned as a pyrrolic nitrogen. They also reported that the amount of XPS peak can be concluded to correspond to the pseudocapacitance. If the assignment is reliable, the work may have a significance in terms of tunable controllability of specific nitrogen species in carbon materials and clarification of the pseudocapacitive active site. However, the conclusion strongly relies on the XPS assignment and the evidence of pyrrolic nitrogen species is currently weak, which does not convince the general reader of Nature Communications. Moreover, the redox peak assignment is ambiguous. Authors did not clearly mention about the "capacitance from oxygen species". Also, the well-known quinone redox is not discussed. The component contribution shown in Fig. 5 should be carefully made by the analysis of the sample before and after the electrochemical measurements. Thus, I recommend the manuscript to be published if the assignment becomes reliable after the major revision. Specific points the authors should consider are listed below:

(1) Assignment of pyrrolic nitrogen

Authors assigned their observed N1s peak at 400.2-400.25 eV as pyrrolic nitrogen. This is well known species and thus normally the assignment is not so problematic. However, in this manuscript, this peak component is a key point of the main conclusion and thus authors should carefully assign it with enough evidence because the other type of nitrogen is known to also show the peak at similar energy position (and the possibility of these other nitrogen species cannot be excluded by the character of precursor because authors conducted the sample treatment for the precursor e.g. heating). Specifically, the peak at 400.2 eV is reported to appear as core level peak of pyridonic nitrogen (Carbon 33, 1641-1653 (1995), Energy Fuels, 8, 896-906 (1994), Fuel, 78, 923-932 (1999), ACS Nano 8, 6856-6862 (2014).). Also, the peak at 400.7 eV is reported as hydrogenated nitrogen species such as hydrogenated pyridinic functionalities (J. Phys. Chem. C120, 29225-29232 (2016).). Therefore, the additional evidence to classify these species should be shown. For example, if the authors show IR spectrum with no correlation of -OH amount and N1s peak intensity, authors can safely avoid the mixing of pyridonic nitrogen. Also, if authors show the drastic decrease of N1s peak by heating at higher temperature, authors may discuss that this is not come from hydrogenated pyridinic nitrogen (because pyrrolic nitrogen is known to be unstable against heating). These are "at least" required evidences, further strong and direct evidences (e.g. STM, TEM, or STEM measurements or other spectroscopic evidences with theoretically calculated models) are required to conclude the main issue of the manuscript and convince the reader of Nature

Communications.

Response:

Thanks for valuable advice from the referee. Indeed, pyridonic nitrogen has similar binding energy with pyrrolic nitrogen in XPS analysis, leading to an extreme difficulty in distinguishing these two nitrogen configurations. As shown in previous reports (Carbon, 1995, 33, 1641-1653; Carbon 1997, 35, 12,1799-1810; Carbon, 1999, 37, 1965-1978; ChemSusChem 2017, 10, 4018-4024), there is the equilibrium of two tautomers, 2-hydroxypyridine and α -pyridone for pyridonic nitrogen-doped carbon materials. (see Supplementary Figure 4b inset.) 2-Hydroxypyridine has a similar chemical environment to pyridine. Herein, 2-hydroxypyridine-type nitrogen has a similar N 1s binding energy to pyridine-type nitrogen. Similar with the nitrogen atom of pyrrole, that of α -pyridone provides two p-electrons to conjugated system, and α -pyridone has a similar chemical environment to pyrrole. Therefore, α -pyridone has a similar N 1s binding energy to pyrrolic nitrogen. It is reasonable that the coexistence of 2-hydroxypyridine and α -pyridone for pyridonic nitrogen-doped carbon materials means that N 1s XPS spectrum of pyridonic nitrogen could be deconvoluted into two peaks at approximately 400.4 eV and at approximately 398.9 eV, generally assigned to pyrrolic nitrogen and pyridinic nitrogen. Therefore, we could conclude that single nitrogen configuration of the binding energy at approximately 400.4 eV for our nitrogen-doped carbon materials is not pyridone. N 1s spectra of nitrogen-doped carbon materials previously reported contains mixed nitrogen configurations approximately 400.4 eV and at approximately 398.9 eV, which means that the accurate assignment of pyrrolic nitrogen and pyridonic nitrogen is extremely difficult for these nitrogen-doped carbon materials.

Furthermore, in order to clarify our single nitrogen configuration as pyrrolic nitrogen or pyridonic nitrogen, 3-fluorophenol-2-hydroxypyridine-formaldehyde resin, namely F-HPF (F, 1/1), was designed and synthesized through similar procedure with 3-fluorophenol-3-aminophenol-formaldehyde resin, F-APF (1/1) in our manuscript. Considering that there are no change of pyridine and adjacent hydroxyl groups in the polymerization of phenol-formaldehyde-like resin does not involve change of pyridine and adjacent hydroxyl groups. In addition, there are no pure pyrrole and pyridine groups for F-APF (1/1) resin. Therefore, F-APF (1/1) resin with merely pyridone groups could be used as model polymer to investigate pyridone group. Therefore, FTIR and XPS spectra of F-APF (1/1) resin were shown in Supplementary Figure 4. FTIR spectrum of F-HPF (F, 1/1) resin shows obvious carbonyl group at 1651 cm^{-1} and aromatic C=C group at 1622 cm^{-1} , revealing the existence of α -pyridone. The survey spectrum of F-HPF (F, 1/1) resin shows the coexistence of these four elements including carbon, nitrogen, oxygen and fluorine. As expected, the N 1s XPS spectrum of F-HPF (F, 1/1) could be deconvoluted into two peaks at 400.35 eV and 398.95 eV, assigned to α -pyridone and 2-hydroxypyridine groups, respectively, which is consistent with the N 1s XPS spectrum of F-HPF (F, 1/1) with carbonyl group of α -pyridone (531.3 eV) and hydroxy group of 2-hydroxypyridine (532.95 eV).

As mentioned above, we could reliably conclude single nitrogen configuration of the binding energy at approximately 400.4 eV for our nitrogen-doped carbon materials is not pyridone group.

Supplementary Figure 4. The FTIR spectra (a) (the inset shows the details of the red dashed box), XPS survey spectrum (b) (the inset is tautomeric schematic of α -pyridone and 2-hydroxypyridine configurations in the resin), high-resolution XPS spectra for N 1s (c) and O 1s (d) of 3-fluorophenol-2-hydroxypyridine-formaldehyde resin, F-HPF (F, 1/1).

Considering that our single nitrogen configuration-doped carbon materials were synthesized through KOH activation, strong basicity largely reduces the probability of hydrogenated pyridinic functionalities with acidity. According to the referee's advice, the SPNCMs (F, 1/1) sample after thermal treatment at 600 °C for 1 h was investigated. Compared with nitrogen content of the SPNCMs (F, 1/1) (4.22 at%), that of the thermally treated SPNCMs (F, 1/1) sample sharply decreases to 2.37 at% based on XPS analysis. Moreover, the N 1s XPS spectrum of (Supplementary Figure R1) shows two peaks at 398.8 eV and 401 eV, corresponding to pyridinic and quaternary nitrogen, respectively, demonstrating the conversion of pyrrolic nitrogen to pyridinic and quaternary nitrogen. Based on the instability of pyrrolic nitrogen against heating, the results confirm the single nitrogen configuration in our carbon materials is not hydrogenated pyridinic functionality.

Supplementary Figure R1. High-resolution N 1s XPS spectrum of SPNCMs (F, 1/1) after thermal treatment at 600 °C for 1 h.

As mentioned above, we could reliably conclude single nitrogen configuration of the binding energy at approximately 400.4 eV for our nitrogen-doped carbon materials is pyrrolic nitrogen.

(2) Redox peak

Authors described about 0.3-0.4 V redox peak as “reversible Faradaic redox reaction of N-5 species and/or oxygen”. It is not clear about Faradaic redox reaction of oxygen because authors did not show any possible exact reaction with evidence (e.g. at least reference should be shown). It seems quinone/hydroquinone redox reaction. In any case, authors should discuss the redox reaction at 0.3-0.4 V more carefully because this is a part of main message of this manuscript.

Response:

Thanks for the valuable reminder from the referee. CV curve of nitrogen-absent CMs (FPF) shows a couple of symmetric redox peaks at 0.3-0.4 V vs SCE, demonstrating reversible Faradaic redox reactions of oxygen species possibly derived from quinone/hydroquinone redox reaction, which is consistent with a couple of symmetric redox peaks at approximately 0.4 V vs SCE for surface-oxidated carbon nanotube (*Journal of The Electrochemical Society*, 2006,153(10), A1823-A1828) and Electrochemically Modified Glassy Carbon (*Journal of The Electrochemical Society*, 2000, 147 (7), 2636-2643). Quinone/hydroquinone redox reaction was described in equation 2:

This result means that high-oxygen-content SPNCMs with similar structures to nitrogen-absent CMs (FPF) could also contribute quinone/hydroquinone redox reaction to pseudocapacitance. Herein, O 1s XPS spectra of our carbon materials including SPNCMs (F, 1/1), SPNCMs (F, 1/3), SPNCMs (F, 1/7) and CMs (FPF) in Supplementary Figure 6 have been added into Supplementary Information. O 1s XPS spectra of all these four carbon materials reveals the presence of hydroxyl/etheric group of the binding energy at 533.1-533.3 eV and carbonyl group of the binding energy at 531.9-532.2 eV. A couple of carbonyl groups could behave as quinone, and a couple of hydroxyl groups behave as hydroquinone. These XPS results, combined with a couple of symmetric redox peaks in CV curve of nitrogen-absent CMs (FPF), confirm that the pseudocapacitance of our single pyrrolic nitrogen-doped carbon materials could partially stem from possible oxygen-dependent quinone/hydroquinone redox reaction.

Supplementary Figure 6. High-resolution O 1s XPS spectra of SPNCMs (F, 1/1) (a), SPNCMs (F, 1/3) (b), SPNCMs (F, 1/7) (c), and CMs (FPF) (d).

(3) Fig. 5

Authors construct Fig. 5 partly based on oxygen content of XPS peak. However, the sample character changes significantly after immersion the sample in electrochemical cell and CV measurements. Thus authors should carefully check the sample condition before and after the measurement otherwise at worse case it causes misleading for general readers of Nature Communications.

Response:

Thanks for the valuable reminder from the referee. Indeed, the character of carbon materials changes after immersing the sample in electrochemical cell and CV measurements. In order to ensure the data reliability based on oxygen content of XPS peak in Fig. 5, the SPNCMs (F, 1/1) samples after charging and discharging at a current density of 1 A g^{-1} in $1.0 \text{ M H}_2\text{SO}_4$ solution were recovered to investigate the change of SPNCMs (F, 1/1) sample. Oxygen content of the fresh SPNCMs (F, 1/1), the recovered SPNCMs (F, 1/1) after charging, and the recovered SPNCMs (F, 1/1) after discharging is 16.69 at.%, 18.64 at.% and 15.45 at.% based on XPS analyses, which means that there is no obvious change in oxygen content for SPNCMs (F, 1/1) before and after GCD test. High-resolution O 1s spectra of these three samples show the conversion of hydroxyl (approximately 533.3 eV) into carbonyl group (approximately 532.1 eV) after charging and corresponding partial reverse conversion after discharging, possibly ascribed to quinone/hydroquinone redox reaction.

Nitrogen content of the fresh SPNCMs (F, 1/1), the recovered SPNCMs (F, 1/1) after charging, and the recovered SPNCMs (F, 1/1) after discharging is 4.22 at.%, 3.97 at.% and 3.69 at.% based on XPS analyses, which means that there is no obvious change in nitrogen content for SPNCMs (F,

1/1) before and after GCD test. High-resolution N 1s spectra of these three samples show the dramatic conversion of pyrrolic nitrogen (approximately 400.2 eV) into oxidized pyrrolic nitrogen (approximately 402 eV) after charging and corresponding obvious reverse conversion after discharging, which confirms pyrrolic nitrogen is an electrochemically active site for nitrogen-doped carbon materials with the pseudocapacitance.

Supplementary Figure R2. High-resolution N 1s (a) and O 1s (b) XPS spectra of SPNCMs (F, 1/1); High-resolution N 1s (a) and O 1s(b) XPS spectra of the recovered SPNCMs (F, 1/1) after charging at a current density of 1 A g^{-1} in $1.0 \text{ M H}_2\text{SO}_4$ solution; High-resolution N 1s (c) and O 1s(d) XPS spectra of the recovered SPNCMs (F, 1/1) after discharging at a current density of 10 A g^{-1} in $1.0 \text{ M H}_2\text{SO}_4$ solution.

(4) Minor point

Authors should show the detail of XPS measurement and analysis conditions more clearly. Specifically, for example, pass energy used for the measurement, peak fitting process (how to determine the width of each peak and what is the peak width), and about charge-up correction (if any).

Response:

Thanks for the referee's advice. XPS analyses were performed with an ESCALAB 250 Xi spectrometer by employing an Al K α X-ray source. The pass energies for XPS survey spectra and high-resolution XPS spectra test of all the samples were 100.0 eV and 30.0 eV, respectively. All the high-resolution XPS spectra were fitted with the Lorentz / Gauss mixing ratios fixed at 0.8, sample charging was corrected by using the C 1s binding energy at 284.8 eV, and Shirley type background subtraction was applied. The FWHMs of N 1s were determined to be approximately 1.8 eV. The FWHMs of O 1s are determined to be approximately 1.8 eV.

Reviewer #3 (Remarks to the Author):

This study reported single pyrrolic N-doped carbon materials (SPNCMs) with a tunable nitrogen content from 0 to 4.22 at.% based on a combined strategy, and inferred that pyrrolic nitrogen species are exactly highly active pseudocapacitive sites for nitrogen-doped carbon materials. The authors try to set an ideal model to understand the contribution of nitrogen species in N-doped carbon materials, offering the principle of materials design for different energy conversions. However, the investigation on the experimental part is very weak (See Below). The quality of this work cannot meet the required standard of Nature commutation. Therefore, the reviewer cannot recommend its acceptance for publication.

Specific comments

1. The chemical structure of NCMs and SPNCMs should be further demonstrated by more physical characterization except XPS.

Response:

Thanks for the advice from the referee. Besides XPS analysis, XAFS analysis of SPNCMs have been mentioned in our initial manuscript. In this revised edition, Raman analyses of NCMs (F, 1/1), SPNCMs (F,1/3), and SPNCMs (F,1/7) and FTIR analyses of F-APF resin (1/1), NCMs (F, 1/1) SPNCMs (F,1/3), and SPNCMs (F,1/7) have been added to investigate their chemical structures. In addition, 3-fluorophenol-2-hydroxypyridine-formaldehyde resin with pyridonic structure has been designed and synthesis to exclude pyridonic nitrogen species in our SPNCMs and to confirm single pyrrolic nitrogen configuration through XPS and FTIR analyses.

Firstly, the chemical structure of SPNCMs (F, 1/1) was demonstrated by XAFS analysis that in the initial manuscript: "XAFS N K edge analysis (Fig. 2f) was conducted to further probe structural nitrogen of SPNCMs (F, 1/1). Besides $1s \rightarrow \sigma^*$ absorption band at 410 eV, there are three well-resolved peaks belonging to $1s \rightarrow \pi^*$ features ranging from 400 to 405 eV. The dominant peak at 400.5 eV, namely N-5, is far stronger than the peaks at 402.4 and 404.2 eV assigned to N-Q (valley-N) and N-X, respectively, which confirms almost single pyrrolic nitrogen configuration in SPNCMs (F, 1/1)." The Raman spectrum of NCMs (F, 1/1) has been added in Fig. 2a, and the corresponding ratio of intensity of D/G bands was determined to be 0.99, slightly lower than that of SPNCMs (F, 1/1) (1.03), possibly because alkaline activation induces more defective sites when nitrogen-containing functionals are completely converted to single pyrrolic

nitrogen configuration. As shown in Figure 2a and Supplementary Figure 9, Raman spectra of SPNCMs (F, 1/1), SPNCMs (F, 1/3), SPNCMs (F, 1/7), and CMs (FPF) show similar feature of sp^2 -hybridized carbon materials with the disorder-induced D-band and a first-order graphite-like G-band. I_D/I_G values of these four carbon materials were determined to be 1.03, 0.96, 0.99, 1.02.

Fig. 2 **a** Raman spectra of NCMs (F, 1/1) and SPNCMs (F, 1/1). **b** XPS survey spectra of F-APF resin (1/1), NCMs (F, 1/1) and SPNCMs (F, 1/1). **c-e** High-resolution N 1s XPS spectra for F-APF resin (1/1) (**c**), NCMs (F, 1/1) (**d**), SPNCMs (F, 1/1) (**e**). **f** N K edge XAFS spectrum of SPNCMs (F, 1/1) (The test was performed in the total electron yield (TEY) mode).

Supplementary Figure 9. Raman spectra of SPNCMs (F, 1/3) (**a**), SPNCMs (F, 1/7) (**b**) and CMs (FPF) (**c**).

Secondly, FTIR spectra in Supplementary Fig. 2 provide some useful information about structural transformation from resin precursor to final SPNCMs. FTIR spectrum of F-APF (1/1) exhibits many characteristic bands including phenol O-H and N-H stretching vibration centered at 3387 cm^{-1} , aromatic C=C stretching at 1623 cm^{-1} , C-F stretching at 1296 cm^{-1} [Journal of Molecular Structure 1009 (2012) 11-15], $-\text{CH}_2-$ asymmetric stretching at 2929 cm^{-1} , symmetric stretching at 2854 cm^{-1} , scissoring vibration at 1477 cm^{-1} , and C-O stretching at 1092 cm^{-1} , verifying successful synthesis of aminophenol-fluorophenol-formaldehyde resin. Compared with FTIR spectrum of F-APF (1/1), that of NCMs (1/1) shows weaker phenol O-H and N-H stretching vibration centered at 3400 cm^{-1} , and relatively weaker C-O stretching at 1092 cm^{-1} , demonstrating

partial removal of oxygen species. In addition, both disappeared C-F stretching vibration at 1296 cm^{-1} and a shift of aromatic C=C stretching to a lower wavenumber (1602 cm^{-1}) verify an occurrence of thermally induced dehalogenation. Compared with FTIR spectrum of NCMs (1/1), that of SPNCMs (F,1/1) shows disappeared aromatic C=C stretching at 1602 cm^{-1} and two newly emerged peaks including C=O stretching vibration at 1637 cm^{-1} and C-O stretching vibration at 1086 cm^{-1} , demonstrating a transition of benzene into carbon structure and simultaneous oxidation of carbon materials to introduce possible phenol, carbonyl, and quinone groups in the KOH activation step.

Supplementary Figure 2. FTIR spectra of F-APF (1/1), NCMs (F,1/1) and SPNCMs (F,1/1)

Thirdly, pyridonic nitrogen has similar binding energy with pyrrolic nitrogen in XPS analysis, leading to an extreme difficulty in distinguishing these two nitrogen configurations. As shown in previous reports (Carbon, 1995, 33, 1641-1653; Carbon 1997, 35, 12,1799-1810; Carbon, 1999, 37, 1965-1978; ChemSusChem 2017, 10, 4018-4024), there is the equilibrium of two tautomers, 2-hydroxypyridine and α -pyridone for pyridonic nitrogen-doped carbon materials. (see Supplementary Figure 4b inset.) 2-Hydroxypyridine has a similar chemical environment to pyridine. Herein, 2-hydroxypyridine-type nitrogen has a similar N 1s binding energy to pyridine-type nitrogen. Similar with the nitrogen atom of pyrrole, that of α -pyridone provides two p-electrons to conjugated system, and α -pyridone has a similar chemical environment to pyrrole. Therefore, α -pyridone has a similar N 1s binding energy to pyrrolic nitrogen. It is reasonable that the coexistence of 2-hydroxypyridine and α -pyridone for pyridonic nitrogen-doped carbon materials means that N 1s XPS spectrum of pyridonic nitrogen could be deconvoluted into two peaks at approximately 400.4 eV and at approximately 398.9 eV, generally assigned to pyrrolic nitrogen and pyridinic nitrogen. Therefore, we could conclude that single nitrogen configuration of the binding energy at approximately 400.4 eV for our nitrogen-doped carbon materials is not pyridone. N 1s spectra of nitrogen-doped carbon materials previously reported contains mixed nitrogen configurations approximately 400.4 eV and at approximately 398.9 eV, which means that the accurate assignment of pyrrolic nitrogen and pyridonic nitrogen is extremely difficult for these nitrogen-doped carbon materials.

Furthermore, in order to clarify our single nitrogen configuration as pyrrolic nitrogen or pyridonic nitrogen, 3-fluorophenol-2-hydroxypyridine-formaldehyde resin, namely F-HPF (1/1), was designed and synthesized through similar procedure with 3-fluorophenol-3-aminophenol-formaldehyde resin, F-HPF (1/1) in our manuscript. Considering that there are no change of pyridine and adjacent hydroxyl groups in the polymerization of phenol-formaldehyde-like resin does not involve change of pyridine and adjacent hydroxyl groups. In addition, there are no pure pyrrole and pyridine groups for F-HPF (1/1) resin. Therefore, F-APF (1/1) resin with merely pyridone groups could be used as model polymer to investigate pyridone group. Therefore, FTIR and XPS spectra of F-HPF (1/1) resin were shown in Supplementary Figure 4. FTIR spectrum of F-HPF (1/1) resin shows obvious carbonyl group at 1651 cm^{-1} and aromatic C=C group at 1622 cm^{-1} , revealing the existence of α -pyridone. The survey spectrum of F-HPF (F, 1/1) resin shows the coexistence of these four elements including carbon, nitrogen, oxygen and fluorine. As expected, the N 1s XPS spectrum of F-HPF (1/1) could be deconvoluted into two peaks at 400.35 eV and 398.95 eV , assigned to α -pyridone and 2-hydroxypyridine groups, respectively, which is consistent with the N 1s XPS spectrum of F-HPF (1/1) with carbonyl group of α -pyridone (531.3 eV) and hydroxy group of 2-hydroxypyridine (532.95 eV).

As mentioned above, we could reliably conclude single nitrogen configuration of the binding energy at approximately 400.4 eV for our nitrogen-doped carbon materials is not pyridone group.

Supplementary Figure 4. The FTIR spectra (a) (the inset shows the details of the red dashed box), XPS survey spectra (b) (the inset is tautomeric schematic of α -pyridonic and 2-hydroxypyridine configurations in the resin), high-resolution XPS spectra for N 1s (c) and O 1s (d) of 3-fluorophenol-2-hydroxypyridine-formaldehyde resin, F-HPF (1/1).

2. As KOH activation was involved in the synthetic procedure, may defect structure and effects should also be considered for the electrochemical performance. It is hard to say the materials developed in this work as ideal model catalysts, particularly after the pyrolysis.

Response:

Thanks for the valuable reminder from the referee. Generally, KOH activation generates nanopores, leading to an increase in defect density (Science, 2011, 332, 1537-1541). Up to date, it is extremely difficult to precisely control and determine the defect type and density of defects for carbon materials. Fortunately, specific surface area represents the density of nanopore-dependent defects to a large content. Moreover, the results of electrochemical surface area (ESA) tests in the non-faradaic voltage range is completely consistent with specific surface areas for SPNCMs (F, 1/1), SPNCMs (F, 1/3), SPNCMs (F, 1/7) and CMs (F, 1/1). Therefore, the capacitance from defects could be approximately regarded as the double layer capacitance.

Up to date, the inevitable nitrogen configuration mixing in carbon materials, greatly limits precisely identifying active nitrogen configurations, fundamentally understanding their mechanisms for specific reactions, and further developing high-performance nitrogen-doped carbon materials for energy storage and conversion. Engineering single nitrogen configuration-doped carbon materials could ideally tackle the problem. Moreover, the defects with different types will be inevitably introduced to nitrogen-doped carbon materials upon their synthesis under present technical conditions. Therefore, in contrast to carbon materials with mixed nitrogen species, our single pyrrolic nitrogen-doped carbon materials with high nitrogen content could be regarded as model nitrogen-doped catalysts. Of course, due to thermal instability of pyrrolic nitrogen configuration, our single pyrrolic nitrogen-doped carbon materials could be used to testify active nitrogen species and to investigate the mechanisms for low-temperature reactions and electrochemical reactions at room temperature. In the revised manuscript, pyrrolic nitrogen was clarified as pseudocapative site based on mutual conversion of pyrrolic nitrogen for SPNCMs (F, 1/1) after charging and discharging (Supplementary Figure R2.).

Supplementary Figure R2. High-resolution N 1s (a) and O 1s(b) XPS spectra of SPNCMs (F, 1/1); High-resolution N 1s (a) and O 1s(b) XPS spectra of the recovered SPNCMs (F, 1/1) after charging at a current density of 1 A g^{-1} in $1.0 \text{ M H}_2\text{SO}_4$ solution; High-resolution N 1s (c) and O 1s(d) XPS spectra of the recovered SPNCMs (F, 1/1) after discharging at a current density of 10 A g^{-1} in $1.0 \text{ M H}_2\text{SO}_4$ solution.

3. Some of the XPS spectra are not well fitted (eg., Figure 2c and Figure 3k). The basic rules of the deconvolution should be obeyed. At least, the half width of the each peak for one element should be consistent!

Response:

According to the referee's valuable advice, we have reanalyzed of all the high-resolution XPS spectra data and replot the corresponding spectra that obeyed the basic rules of the deconvolution. XPS analyses were performed with an ESCALAB 250 Xi spectrometer by employing an Al K α X-ray source. The pass energies for XPS survey spectra and high-resolution XPS spectra test of all the samples were 100.0 eV and 30.0 eV, respectively. All the high-resolution XPS spectra were fitted with the Lorentz / Gauss mixing ratios fixed at 0.8, sample charging was corrected by using the C 1s binding energy at 284.8 eV, and Shirley type background subtraction was applied. The

FWHMs of N 1s were determined to be approximately 1.8 eV. The FWHMs of O 1s are determined to be approximately 1.8 eV.

4. Page 2, Line 52. “The authors stated that “it seems to be an impossible mission to engineer single nitrogen configuration-doped carbon materials due to an inevitable thermal treatment involved in synthesizing almost all the nitrogen-doped carbon materials primarily via either post-synthetic or in situ doping methods 5,9,23.” However, Ref. 21 has already reported the synthesis of pure pyridinic N or graphitic N doped HOPG. Also, the Ref. 23 reported on nanoporous carbon spheres, rather than doping type of N.

Response:

According to the referee’ reminder, we have carefully reread Ref. 21 (Page 361, *Science* **351**, 361-365 (2016)) in our initial manuscript.

In Ref. 21 (Page 361, *Science* **351**, 361-365 (2016)), the authors described as follows: “To determine the active site conclusively, we develop four types of model catalysts with well-defined p-conjugation based on highly oriented pyrolytic graphite (HOPG): (i) pyridinic N-dominated HOPG (pyri-HOPG); (ii) graphitic N-dominated HOPG (grap-HOPG), and for comparison, (iii) edges patterned on the surface without N (edge-HOPG); and (iv) clean-HOPG (see supplementary methods and fig. S1).” Pyridinic N-dominated HOPG (pyri-HOPG) and graphitic N-dominated HOPG (grap-HOPG) mentioned by the authors don’t refer to *pure pyridinic N or graphitic N doped HOPG*. As shown in the inset of Fig. 1F in Ref. 21, the percentage of graphitic N in graphitic N-dominated HOPG is about 82.2 % (0.60/0.73) and the percentage of pyridinic N in pyridinic N-dominated HOPG is about 95 % (0.57/0.60).

Ref. 21. Fig. 1. Structural and elemental characterization of four types of N-HOPG model catalysts and their ORR performance. (E) N 1s XPS spectra of model catalysts. (F) ORR results for model catalysts corresponding to (E). Nitrogen contents of the model catalysts are shown as the inset in (F).

Thanks for valuable advice of the referee. The chapter named “Doping and incorporation of heteroatoms into NCS.” (Ref. 23, page 768, line 9, *Nat. Mater.* 14, 763-774 (2015)) describes the main methods to introduce nitrogen into nanoporous carbon spheres (NCS). However, as pointed by the referee, this review focuses more on nanoporous carbon spheres than on doping N atoms in carbon materials. Thus, we have removed this reference from our manuscript.

5. Some of the previous literature reported that pyridinic (pyrrolic also) N species can be transferred into the graphitic N during the pyrolysis, and that the pyrrolic N is not stable during the electrochemical catalytic process, how about the stability of the catalyst developed in this work?

Response:

Thanks for the valuable reminder from the referee. Indeed, pyridinic (pyrrolic also) N species can be transferred into the graphitic N during the pyrolysis with the temperature higher than 800 °C (Carbon 37 (1999) 1965 -1978). Therefore, our single pyrrolic nitrogen-doped carbon materials were synthesized at 500 °C to avoid the conversion of pyrrolic nitrogen to pyridinic nitrogen, and to graphitic nitrogen.

According to the referee’s advice, electrochemical stability of SPNCMs (F, 1/1) electrode material was evaluated. As shown in Supplementary Figure 13a, SPNCMs (F, 1/1) preserves high specific capacitance of 467 F g⁻¹ up to 10 A g⁻¹ and reveals ultrahigh stability over 30,000 cycles during the GCD tests at the large current density of 10 A g⁻¹. In order to confirm the electrochemical stability of SPNCMs (F, 1/1), SEM, TEM, HRTEM, and XPS analysis of the recovered SPNCMs (F,1/1) electrode material after GCD test over 30,000 cycles were performed. SEM and TEM images demonstrate that the recovered SPNCMs (F,1/1) electrode material preserves the initial morphologic feature of randomly assembled layers. As observed in HRTEM image, the recovered SPNCMs (F,1/1) electrode material preserves HRTEM image exhibits similar turbostratic carbon structure to fresh SPNCMs (F,1/1) material. EDS analysis confirms the recovered SPNCMs (F, 1/1) electrode material maintains uniform doping of nitrogen species into the turbostratic carbon framework (Supplementary Figure 13). Furthermore, XPS analysis reveals that the recovered SPNCMs (F,1/1) electrode material preserves similar nitrogen content (4.8 at.%) to the fresh SPNCMs (F,1/1) material (4.22 at.%). Different from the single pyrrolic nitrogen configuration of the fresh SPNCMs (F,1/1) material (Fig. 2e), the N 1s XPS spectrum of the recovered SPNCMs (F,1/1) electrode material shows the coexistence of pyrrolic nitrogen at 400.4 eV and oxidized pyrrolic nitrogen at 401.9 eV, which could be ascribed to that there is the redox reaction of pyrrolic nitrogen/electrochemically oxidized pyrrolic nitrogen upon charging-discharging cycles, and rapid charging-discharging cycles at 10 A g⁻¹ of the large current density possibly results in incomplete use of pyrrolic nitrogen and incomplete reduction of

electrochemically oxidized pyrrolic nitrogen. The O 1s XPS spectrum of the recovered SPNCMs (F,1/1) electrode material still shows the coexistence of hydroxyl/etheric group of the binding energy at 533.3 eV and carbonyl group of the binding energy at 532 eV, similar to O 1s XPS spectrum of the fresh SPNCMs (F,1/1) material. The results mentioned above confirm high stability of the SPNCMs (F,1/1) electrode material.

Supplementary Figure 13. Gravimetric capacitance (a) and capacity retention (%) (b) of the SPNCMs (F, 1/1) electrodes in 1.0 M H_2SO_4 solution in a three-electrode system at a charge current of $10 A g^{-1}$ over 30,000 cycles, SEM (a), TEM (b), HRTEM (c), HAADF-STEM (d) images, the elemental carbon (d) and nitrogen (f) mapping images, high-resolution XPS spectra for N 1s (c) and O 1s (d) of the recovered SPNCMs (F, 1/1) electrodes after aforementioned GCD test above over 30,000 cycles.

6. The authors stated that they reported one kind of single pyrrolic N-doped carbon materials (SPNCMs) as ideal model materials for supercapacitor. However, the XPS results show a high content of oxygen, and Raman spectra show the presence of numerous defects. Both of the oxygen functional groups and defects will affect the (pseudo-) capacitance, which should be distinguished from the contribution of the pyrrolic N.

Response:

Generally, KOH activation generates nanopores, leading to an increase in defect density (Science, 2011, 332, 1537-1541). Up to date, it is extremely difficult to precisely control and determine the defect type and density of defects for carbon materials. Fortunately, specific surface area represents the density of nanopore-dependent defects to a large content. Moreover, the results of electrochemical surface area (ESA) tests (Supplementary Figure 14) in the non-faradaic voltage range is completely consistent with specific surface areas for SPNCMs (F, 1/1), SPNCMs (F, 1/3), SPNCMs (F, 1/7) and CMs (F, 1/1) in Supplementary Figure 7. Therefore, the capacitance from defects was regarded as the double layer capacitance in our manuscript. To further explore the N-5 pseudocapacitive contribution, based on these nearly rectangular CV curves (Supplementary Fig. 14a-d) ranging from 0 to 0.05 V vs. SCE and the slopes of the scan rate-dependent current density curves (Supplementary Fig. 14e), EDLCs of SPNCMs (F, 1/1), SPNCMs (F, 1/3), SPNCMs (F, 1/7), and CMs(FPF) were firstly calculated though ESA tests to be about 125, 263, 211, and 183 F g⁻¹, respectively, highly consistent with their corresponding specific surface areas (SSA) of 450, 950, 873, and 880 cm² g⁻¹, respectively. Based on their EDLCs and total specific capacitance of 584.8, 486.7, 412.4, and 301.5 F g⁻¹ at 1 mV s⁻¹, respectively, their corresponding faradaic pseudocapacitances can be calculated to be 459.8, 223.7, 201.4 and 118.5 F g⁻¹, respectively.

CV curve of nitrogen-absent CMs (FPF) shows a couple of symmetric redox peaks at 0.3-0.4 V vs SCE, demonstrating reversible Faradaic redox reactions of oxygen species possibly derived from quinone/hydroquinone redox reaction, which is consistent with a couple of symmetric redox peaks at approximately 0.4 V vs SCE for surface-oxidized carbon nanotube (*Journal of The Electrochemical Society*, 2006,153(10), A1823-A1828) and Electrochemically Modified Glassy Carbon (*Journal of The Electrochemical Society*, 2000, 147 (7), 2636-2643). Quinone/hydroquinone redox reaction was described in equation 2:

This result means that high-oxygen-content SPNCMs with similar structures to nitrogen-absent CMs (FPF) could also contribute quinone/hydroquinone redox reaction to pseudocapitance. Herein, O 1s XPS spectra of our carbon materials including SPNCMs (F, 1/1), SPNCMs (F, 1/3), SPNCMs (F, 1/7) and CMs (FPF) in Supplementary Figure 6 have been added into Supplementary Information. O 1s XPS spectra of all these four carbon materials reveals the presence of hydroxyl/etheric group of the binding energy at 533.1-533.3 eV and carbonyl group of the binding energy at 531.9-532.2 eV. A couple of carbonyl groups could behave as quinone, and a couple of hydroxyl groups behave as hydroquinone. These XPS results, combined with a couple of symmetric redox peaks in CV curve of nitrogen-absent CMs (FPF), confirm that the pseudocapitance of our single pyrrolic nitrogen-doped carbon materials could partially stem from possible oxygen-dependent quinone/hydroquinone redox reaction.

So far, due to great difficulty to precisely determine the structure and number of electrochemically active oxygen species and to exclude electrochemically inert oxygen species for carbon materials, the merely oxygen-dependent pseudocapacitance of CMs (FPF) with symmetric redox lumps in their CV measurement at different scan rates (Supplementary Fig. 10d) has to be assumed to derive from all oxygen atoms with identical pseudocapacitive contribution. Considering the pseudocapacitance of 118.5 F g^{-1} for CMs (FPF) and oxygen content determined by XPS analysis (Supplementary Table 2), the specific pseudocapacitance per mole of oxygen atom can be calculated to be 8471.75 F . Accordingly, the oxygen-dependent specific pseudocapacitances were calculated to be 105.9, 100.8, and 146.3 F g^{-1} for SPNCMs (F, 1/1), SPNCMs (F, 1/3), and SPNCMs (F, 1/7), respectively, and then their corresponding single pyrrolic nitrogen-dependent pseudocapacitances can be calculated to $353.9, 122.9, 55.1 \text{ F g}^{-1}$, respectively (Supplementary Table 3, Fig. 5).

Therefore, the pseudo-capacitances of oxygen functional groups and the defects-dependent double layer capacitance were subtracted from the total capacitance, and the rest capacitance could attribute to single pyrrolic nitrogen-dependent pseudocapacitance. Finally, it could be deduced that the pseudocapacitance is positively dependent on N-5 content in nitrogen-doped carbon materials, thanks to our SPNCMs with both similar ion diffusion capacity, electron conductivity and single pyrrolic nitrogen configuration.

Supplementary Figure 7. N₂ adsorption/desorption analysis of SPNCMs (F, 1/1), SPNCMs (F, 1/3), SPNCMs (F, 1/7), and CMs (FPF). (a) High-resolution, low-pressure N₂ (77.5 K) isotherms. (b) Cumulative pore volume and (c, d) pore-size distribution for N₂ adsorption. The cumulative pore volume (b) and pore size distribution (c, d) curves of these four carbon materials were plotted according to their corresponding nitrogen adsorption curves by an Original Density Functional Theory (ODFT) method.

Supplementary Figure 10. CV tests of SPNCMs (F, 1/1) (a), SPNCMs (F, 1/3) (b), SPNCMs (F, 1/7) (c), and CMs (FPF) (d) at different scan rates from 1 to 20 mV s⁻¹ in 1.0 M H₂SO₄ solution in the potential window of -0.15 to 0.85 V vs. SCE. (e) The corresponding specific capacitance values of the above different carbon materials at different scan rates.

Supplementary Figure 14. Measurement of EDLC of different carbon materials by electrochemical surface area (ESA) without Faradic reaction in the voltage range (0-0.05 V vs. SCE). CV curves of SPNCMs (F, 1/1) (a), SPNCMs (F, 1/3) (b), SPNCMs (F, 1/7) (c), and CMs (FPF) (d) electrodes measured in 1 M H₂SO₄ at scan rates from 0.5 to 3.0 mV s⁻¹; (e) the corresponding ESA values determined by the linear curves of the discharge current density at 0.01 V (vs. SCE) vs. the scan rate. The capacitance of SPNCMs (F, 1/1) (0.493 mg cm⁻²), SPNCMs (F, 1/3) (0.3825 mg cm⁻²), SPNCMs (F, 1/7) (0.255 mg cm⁻²) and CMs (FPF) (0.595 mg cm⁻²) were measured to be 62, 100, 54 and 109 mF cm⁻², the corresponding EDLCs were about 125, 263, 211, 183 F g⁻¹, respectively.

Supplementary Figure 6. High-resolution O 1s XPS spectra of SPNCMs (F, 1/1) (a), SPNCMs (F, 1/3) (b), SPNCMs (F, 1/7) (c), and CMs (FPF) (d).

Fig. 5 The variations of pseudocapacitances with pyrrolic nitrogen content in CMs (FPF), SPNCMs (F, 1/7), SPNCMs (F, 1/3), and SPNCMs (F, 1/1).

Supplementary Table 2. C, N, O and F content of the samples determined by XPS analysis

Samples	C (at.%)	N (at. %)	O (at.%)	F (at.%)
F-APF (1/1)	75.98	4.27	13.59	6.17
NCMs (F, 1/1)	85.88	4.36	8.83	0.93
SPNCMs (F, 1/1)	78.4	4.22	16.69	0.7
SPNCMs (F, 1/3)	81.52	3.07	15.09	0.31
SPNCMs (F, 1/7)	76.17	1.51	22.32	0
CMs (FPF)	82.22	0	17.78	0

Supplementary Table 3. EDLC and pseudo-capacitive contributions to the total specific capacity of SPNCMs (F, 1/1), SPNCMs (F, 1/3), SPNCMs (F, 1/7), and CMs (FPF) in three-electrode system at 1 mV s^{-1} in $1.0 \text{ M H}_2\text{SO}_4$ electrolyte

Samples	SSA ($\text{m}^2 \text{ g}^{-1}$)	Pyrrolic N (at.%)	Specific capacity (F g^{-1})	EDLC (F g^{-1})	Faradaic capacitance (F g^{-1})	
					Faradaic capacitance from oxygen species	Faradaic capacitance from pyrrolic nitrogen
SPNCMs (F, 1/1)	450	4.22	584.8	125	105.9	353.9
SPNCMs (F, 1/3)	950	3.07	486.7	263	100.8	122.9
SPNCMs (F, 1/7)	873	1.51	412.4	211	146.3	55.1
CMs (FPF)	880	0	301.5	183	118.5	0

7. The graphitization degree should also be determined (e.g. by XRD), which may show a relatively low graphitization degree with respect to the HOPG doped by pyrrolic N only previously-reported by Nakamura et al. (i.e., Ref. 21). The so-called single pyrrolic N-doped carbon (SPNCM) is inferior to its HOPG counterparts.

Response:

Thanks for the referee's valuable advice. As shown in Supplementary Figure 1a, the interplanar spacing of the (002) plane of small microcrystalline graphite of SPNCMs (F, 1/1) was calculated to be 0.3645 nm , far larger than 0.3354 nm , interplanar spacing of the (002) plane of ideal graphite, which indeed means the relatively low graphitization degree near to 0% for SPNCMs (F, 1/1). Phenol-formaldehyde resin belonging to glassy carbon precursor are difficult to graphitize, and even glassy carbon (Type I) pursued from Alfa Aesar shows 0.3694 nm of the interplanar spacing of the (002) plane of small microcrystalline graphite and relatively low graphitic degree near to 0%, according to XRD spectrum of glassy carbon (Supplementary Figure R3.). Glassy carbon with good electroconductivity is widely used as an electrode material in electrochemistry.

Compared with the HOPG doped by pyrrolic N, our SPNCMs show lower graphitic degree. However, there are mixed nitrogen configurations for HOPG doped by pyrrolic N according to previous reports such as *Ref. 21* in our initial manuscript, suppressing precise recognition of electrochemically active nitrogen species and reliable investigation of corresponding electrochemical mechanism of pyrrolic nitrogen. Our SPNCMs could tackle the problem mentioned above. Indeed, the single pyrrolic N-doped carbon (SPNCM) is inferior to its HOPG counterparts. However, there is no any report on single pyrrolic nitrogen-doped HOPG up to date, due to uncontrolled nitrogen doping for HOPG. Single pyrrolic nitrogen-doped carbon materials with high graphitic degree is still a great challenge and our future focus.

Supplementary Figure 1. (a) XRD spectra of F-APF (1/1), NCMs (F,1/1) and SPNCMs (F,1/1); SEM (b), TEM (c), HRTEM (d), HAADF-STEM (e) images and the elemental mapping images (carbon (f), nitrogen (g)) of SPNCMs (F,1/1).

Supplementary Figure R3. XRD spectrum of glassy carbon (Type I)

8. In order to claim the supercapacitance from the single pyrrolic N-doped carbon materials (SPNCMs), the single pyridinic N-doped, and single graphitic N-doped SPNCM counterparts need to be prepared as references so that the oxygen and defect effects can be neglected.

Response:

Thanks for the referee's advice. Unfortunately, there is no any reliable strategy to construct the single pyridinic N-doped, and single graphitic N-doped carbon materials. Even though these two carbon materials can be successfully synthesized, different type and density of oxygen species and defects may be combined into these two carbon materials, leading to that the oxygen and defect

effects can be neglected. Moreover, the single pyridinic N-doped, and single graphitic N-doped carbon counterparts may only clarify the pseudocapacitances from pyridinic and graphitic nitrogen, respectively, not claim the pseudocapacitance from pyrrolic nitrogen for single pyrrolic nitrogen-doped carbon materials.

9. Some Refs cited in this study is not suitable, for example:

Page 2, Line 33. "...oxygen reduction reaction (ORR)6-9, CO2 reduction reaction10-12, oxygen evolution reaction (OER)13-15, hydrogen evolution reaction (HER)16, nitrogen reduction reaction17-20, and so on...", however, the Ref. 8 focused on "Fe-Nx" active sites for ORR, while Ref. 17 studied on the catalysis of CO2 reduction.

Response:

Thanks for the referee's valuable advice. We have carefully checked the references cited in the manuscript. Related references have been revised. Ref. 8 has been deleted, and Ref. 17 has transferred to supply CO₂ reduction reaction.

Extra revision.

We are apologized to some minor errors.

1. In Line 14 Page 15 in the revised manuscript, the data "301.5 F g⁻¹" was mistakenly written as "118.5 F g⁻¹" due to typo error. This wrong writing has been revised.
2. In Page 6 in the revised Supplementary information, according to the advice from the referees, all high-resolution N 1s XPS spectra have been refitted, and the relative nitrogen configuration ratios in the samples were replotted in Supplementary Figure 5.
3. In Line 11 Page 15 in the revised manuscript, the data "183 F g⁻¹" was mistakenly written as "182 F g⁻¹" due to typo error. This wrong writing has been revised.
4. Considering that the key contribution to synthesize and characterize model resin, 3-fluorophenol-2-hydroxypyridine-formaldehyde resin to determine nitrogen species in our carbon materials from Wei Yang at Yanshan University and the key contribution to characterize our carbon materials from Wei Li at Beijing University with respect to the revised manuscript, we have added Wei Yang and Wei Li as co-authors in our revised manuscript.
5. Considering the help from Prof. Zhisheng Zhao at Yanshan University for the XRD analysis of our nitrogen-doped carbon materials in our revised manuscript. The contribution of Prof. Zhisheng Zhao has been added into Acknowledgements in our revised manuscript.

Reviewers' Comments:

Reviewer #1:

Remarks to the Author:

The authors have addressed all my concerns. Therefore, I would like to recommend its publication in Nature Communications.

Reviewer #2:

Remarks to the Author:

Unfortunately the revision is not satisfactory in terms of the following points:

(1) XPS analysis

I have mentioned in my previous review that the authors should show the width of XPS peak used for fitting. The revision is not enough. This is an important point because, for example, general reader can easily doubt the deconvolution analysis in Fig. 3 if the different width is used for the fitting analysis. In fact, it seems not fair in the current version because Figs. b-n are fitted with multiple peaks while Figs. d-p were fitted by single peak (there is no clear explanation about this delicate point). That is, Figs. d-p can be probably fitted with multiple peaks! If so, the conclusion is totally different!

Moreover, in the reply letter to the third reviewer, authors described as

"In Ref. 21 (Page 361, Science 351, 361-365 (2016)), the authors described as follows: "To determine the active site conclusively, we develop four types of model catalysts with well-defined p-conjugation based on highly oriented pyrolytic graphite (HOPG): (i) pyridinic N-dominated HOPG (pyri-HOPG); (ii) graphitic N-dominated HOPG (grap-HOPG), and for comparison, (iii) edges patterned on the surface without N (edge-HOPG); and (iv) clean-HOPG (see supplementary methods and fig. S1)." Pyridinic N-dominated HOPG (pyri-HOPG) and graphitic N-dominated HOPG (grap-HOPG) mentioned by the authors don't refer to pure pyridinic N or graphitic N doped HOPG. As shown in the inset of Fig. 1F in Ref. 21, the percentage of graphitic N in graphitic N-dominated HOPG is about 82.2 % (0.60/0.73) and the percentage of pyridinic N in pyridinic N-dominated HOPG is about 95 % (0.57/0.60)."

But the indicated figure extracted from (Science 351, 361-365 (2016)) in the reply letter by the authors show higher quality compared to the result by the authors in terms of selectivity of the peak component in the material. The derived percentage (pointed out by the authors) is just a result due to the fitting accuracy (I doubt poor fitting analysis by the authors now because there is no clear description about this important analysis). In any case, the general reader cannot convince that the XPS results in Figs. 3d-p are composed by a single component.

Moreover, there are other reports showing single component nitrogen doping (for example, J. Mater. Chem., 2011, 21, 8038).

I thus suggest that authors should revise the manuscript as originally the third referee pointed out.

(2) Pyridonic nitrogen and hydrogenated pyridinic nitrogen

Authors added the description about pyridonic nitrogen to support the assignment for the pyrrolic nitrogen. The discussion is good. It is however not strong evidence but possible interpretation. The strong experimental evidences to support the assignment is still required, strictly to say.

Moreover, authors did not revise the manuscript for the hydrogenated pyridinic nitrogen though some descriptions are there in the reply letter.

(3) "Significant figures" and instrumental calibration

In the revision, authors describe as "Therefore, interplanar spacing of the (002) plane of small microcrystalline graphite of SPNCMs (F, 1/1) was calculated to be 0.3645 nm, far larger than 0.3354 nm, interplanar spacing of the (002) plane of ideal graphite."

But there is no description about the accuracy about the value 0.3645 nm. In the previous version of the manuscript, the value was very rough. The authors should clearly describe the reliable "Significant figures". Also, authors should clearly describe how to calibrate the instrumental deviation of the measured distances from XRD, TEM and STEM from the exact value.

Reviewer #3:

Remarks to the Author:

Although the authors have added much data in the revised manuscript, the Reviewer still insists that the quality of the article is the most important criterion for the publication in prestigious journals, like Nature Commutation. Thus, the reviewer rejects acceptance of the revised manuscript for publication.

Specific comments:

1. Many of the results are improperly interpreted. First of all, the authors demonstrated the defects in the porous carbon mainly by Raman, for example: the author stated that "The Raman spectrum of NCMs (F, 1/1) has been added in Fig. 2a, and the corresponding ratio of intensity of D/G bands was determined to be 0.99, slightly lower than that of SPNCMs (F, 1/1) (1.03), possibly because alkaline activation induces more defective sites when nitrogen-containing functionals are completely converted to single pyrrolic nitrogen configuration. As shown in Figure 2a and Supplementary Figure 9, Raman spectra of SPNCMs (F, 1/1), SPNCMs (F, 1/3), SPNCMs (F, 1/7), and CMs (FPF) show similar feature of sp²-hybridized carbon materials with the disorder-induced D-band and a first-order graphite-like G-band. ID/IG values of these four carbon materials were determined to be 1.03, 0.96, 0.99, 1.02." and "Similar to the Raman spectrum of NCMs (F, 1/1), that of SPNCMs (F, 1/1) (Fig. 2a) shows typical feature of sp²-hybridized carbon materials with abundant defects induced mainly by nitrogen doping, as inferred by the high intensity ratio of about 1.03 between the disorder-induced D-band at 1353 cm⁻¹ and a first-order graphite-like G-band at 1593 cm⁻¹ (ID/IG) of SPNCMs (F, 1/1)51."

However, Raman D band does not in any sense whatsoever represent disorder carbons, though many people have published so. The authors should cite proper Raman papers on special Raman spectroscopy of graphene or carbon materials, not materials papers with often wrong interpretations of fundamental Raman results. Check work by Andrea Ferrari, Denis Basko, Mildred Dresselhaus, Ado Jorio, Riichiro Saito, Kentaro Sato, or any other systematic work on Raman of graphene, defective graphene, amorphous carbon, defects and impurities in graphene-like materials.

The above interpretation is only acceptable for well-ordered graphene. In fact, it is the opposite for highly defective graphitic carbon. The Tunistra-Koenig (T-K) relation breaks down, and the explanation of high D/G peak corresponding to high defect is not often correct. In addition, assigning D/G ratio without a systematic fitting can be meaningless. A comprehensive and appropriate Raman analysis should be carried out to substantiate structural claims. What's more, it's better to present a statistical analysis of the Raman D/G peak ratios.

In this study, the D peak of the typical sample is too wide, and the diffraction peaks of (002) and (100) are broad, this kind of carbon just like amorphous carbon, rather than sp²-conjugated carbon! At least, the 2D band (the value should be labeled) in the Raman spectra and SAED should be provided for demonstrating the so-called sp²-conjugated structure!

The authors stated that "The ratio of intensity of D/G bands of NCMs (F, 1/1) was determined to be 0.99, slightly lower than that of SPNCMs (F, 1/1) (1.03), possibly because alkaline activation induces more defective sites," However, they also described that Raman spectra of SPNCMs (F, 1/1), SPNCMs (F, 1/3), SPNCMs (F, 1/7), and CMs (FPF) with the ID/IG values of 1.03, 0.96, 0.99, 1.02 show similar feature of sp²-hybridized carbon materials with the disorder-induced D-band and a first-order graphite-like G-band. It is inappropriate to interpret the data according to their subjective needs!

2. Page 6, Line 19. "Additionally, EDS analysis shows uniform doping of nitrogen species into the

turbostratic carbon framework for SPNCMs (F, 1/1) (Supplementary Fig. 1f, g).” Firstly. Because the elements of N and C are two neighbor light elements, it’s very hard to distinguish them from each other by this kind of element mapping, authors should reconsider it. Also, the shapes of the images in Supplementary Fig.s 1e, f, g are different!

3. The reviewer doesn’t think the structures of NCMs and SPNCMs shown in Figure 1 are appropriate, as the O contents of the samples are not low!

4. In the Reviewer’s opinion, the typical sample in this study is N and O co-doped carbon, however, the authors didn’t take the effect of O on the dopant N into account at all!

5. The authors stated that “thermal treatment below about 600 °C always yields lowly graphitized nitrogen-doped carbon materials with poor electrical conductivity...” What are the specific values of conductivity of the samples in this study, how about these values compared with commercial conductive carbon?

6. The topic of this work is “sp²-conjugated Carbon Materials”, why the authors cited 16 of Ref.s on sp³ and sp carbon in the introduction section in the revised version? So much Ref.s (1/4 of the total) make the reviewer a little confused!

7. For HR-XPS spectra, the basic rules of the deconvolution should be obeyed. At least, the half-width of each peak for one element should be consistent! There are still some problems of the deconvolution in Figure 3f, Supplementary Figure 6c, and Supplementary Figure 12c in the revised version!

Reviewers' comments:

Reviewer #1 (Remarks to the Author):

The authors have addressed all my concerns. Therefore, I would like to recommend its publication in Nature Communications.

Thanks for your kind review for our manuscript.

Reviewer #2 (Remarks to the Author):

Q1. XPS analysis

I have mentioned in my previous review that the authors should show the width of XPS peak used for fitting. The revision is not enough. This is an important point because, for example, general reader can easily doubt the deconvolution analysis in Fig. 3 if the different width is used for the fitting analysis. In fact, it seems not fair in the current version because Figs. b-n are fitted with multiple peaks while Figs. d-p were fitted by single peak (there is no clear explanation about this delicate point). That is, Figs. d-p can be probably fitted with multiple peaks! If so, the conclusion is totally different!

Response:

Thanks for valuable advice from the referee. We have carefully checked our XPS data and deconvolution analyses. According to literatures[*Carbon* 33, 1021-1027 (1995); *Fuel* 88, 1871-1877 (2009)], there is no significant difference of the N 1s binding energies in primary amine, secondary amine and tertiary amine groups, therefore, these N 1s peaks could not be fitted with multiple peaks but should be fitted by single peak. In our previous version, multiple peaks-fitting were wrongly used, only considering the different chemical states of these N configurations and not considering the resolution of XPS measurement, which directly lead to smaller FWHMs of these N 1s peaks than FWHM of N 1s peak of pyrrolic nitrogen. In the current version of manuscript, the FWHM of N 1s for secondary amine and tertiary amine groups in polymer precursors fitted by single peak are 1.9, 1.9, 1.9, 2.0 and 1.9 eV, respectively, as shown in Figure 2c, Figure 3b, f, j, k n and Supplementary Table 2. And at the same time, the FWHMs of N 1s for pyrrolic N configuration in our samples were fixed at ~2.0 eV with single peak.

All the FWHM values for N 1s in the samples were listed in the following tables.

The analysis of refitted XPS peaks with the same width of FWHM approximately 1.9 eV for N 1s and approximately 1.8 eV for O 1s for our carbon materials confirms the existence of single pyrrolic nitrogen configuration in our carbon materials.

Fig. 2 **a** Normalized Raman spectra of SPNCMs (F, 1/1). **b** XPS survey spectra of F-APF resin (1/1), NCMs (F, 1/1) and SPNCMs (F, 1/1). **c-e** High-resolution N 1s XPS spectra for F-APF resin (1/1) (**c**), NCMs (F, 1/1) (**d**), SPNCMs (F, 1/1) (**e**). **f** N K edge XAFS spectrum of SPNCMs (F, 1/1) (The test was performed in the total electron yield (TEY) mode).

Fig. 3 **a, e, i, m** XPS survey spectra of samples. High-resolution N 1s XPS spectra for F-APF resin (1/3) (**b**), NCMs (F, 1/3) (**c**), SPNCMs (F, 1/3) (**d**), F-APF resin (1/7) (**f**), NCMs (F, 1/7) (**g**), SPNCMs (F, 1/7) (**h**), F-APF resin (Cl) (**j**), NCMs (Cl) (**k**), SPNCMs (Cl) (**l**), F-APF resin (Br) (**n**), NCMs (Br) (**o**), and SPNCMs (Cl) (**p**).

Supplementary Table 2. Binding energies and FWHM for N1s in Figure 2 and Figure 3

Sample	-NH-, >N-		Pyrolic N		Figure
	B.E.(eV)	FWHM	B.E.(eV)	FWHM	
F-APF (1/1)	399.25	1.9	-	-	Figure 2c
NCMs (F, 1/1)	398.9	1.9	400.3	2	Figure 2d
SPNCMs (F, 1/1)	-	-	400.25	2	Figure 2e
F-APF (1/3)	399.53	1.9	-	-	Figure 3b

NCMs (F, 1/3)	398.97	1.9	400.57	2	Figure 3c
SPNCMs (F, 1/3)	-	-	400.25	2	Figure 3d
F-APF (1/7)	399.55	1.9	-	-	Figure 3f
NCMs (F, 1/7)	399	1.9	400.4	2	Figure 3g
SPNCMs (F, 1/7)	-	-	400.35	2	Figure 3h
Cl-APF	399.5	2	-	-	Figure 3j
NCMs (Cl)	398.9	1.9	400.3	2	Figure 3k
SPNCMs (Cl)	-	-	400.35	2	Figure 3l
Br-APF	399.5	1.9	-	-	Figure 3n
NCMs (Br)	398.9	1.9	400.5	2	Figure 3o
SPNCMs (Br)	-	-	400.4	2	Figure 3p

Supplementary Table 3. Binding energies and FWHM for N1s of NCMs (APF) and NCMs (F, 3/1) in Supplementary Figure 3

Sample	NCMs (APF)		NCMs (F, 3/1)	
	B.E.(eV)	FWHM	B.E.(eV)	FWHM
NCMs (APF)	400	1.9		
NCMs (F, 3/1)			401.7	2

Supplementary Table 4. Binding energies and FWHM for N1s and O1s of 3-fluorophenol-2-hydroxypyridine-formaldehyde resin, F-HPF (F, 1/1) in Supplementary Figure 4

Sample	Pyridinic N		Pyridonic N		-C=O		-O-H, C-O-C	
	B.E.(eV)	FWHM	B.E.(eV)	FWHM	B.E.(eV)	FWHM	B.E.(eV)	FWHM
F-HPF (F, 1/1)	398.95	1.9	400.35	1.9	531.3	1.8	532.95	1.8

Supplementary Table 5. Binding energies and FWHM for O1s in Supplementary Figure 6

Sample	-C=O		-OH, C-O-C	
	B.E.(eV)	FWHM	B.E.(eV)	FWHM
SPNCMs (F, 1/1)	532	1.85	533.5	1.9
SPNCMs (F, 1/3)	531.9	1.8	533.3	1.8
SPNCMs (F, 1/7)	532.8	1.85	533.7	1.9
CMs (FPF)	532.1	1.8	533.3	1.8

Supplementary Table 6. Binding energies and FWHM for N1s and O1s of SPNCMs (F, 1/1) in Supplementary Figure 12

Sample	Pyrrolic N		Electrochemically oxidized pyrrolic N		-C=O		-O-H, C-O-C	
	B.E.(eV)	FWHM	B.E.(eV)	FWHM	B.E.(eV)	FWHM	B.E.(eV)	FWHM
After charging	400.2	2	402	2	532.1	1.8	533.3	1.8
After	400.2	2	401.8	2	532.2	1.8	533.2	1.85

discharging

Supplementary Table 7. Binding energies and FWHM for N1s and O1s of SPNCMs (F, 1/1) in Supplementary Figure 13

Sample	Pyrrolic N		Electrochemically oxidized pyrrolic N		-C=O		-O-H, C-O-C	
	B.E.(eV)	FWHM	B.E.(eV)	FWHM	B.E.(eV)	FWHM	B.E.(eV)	FWHM
After 30,000 cycles	400.4	2	401.8	2	532	1.8	533.25	1.8

Moreover, in the reply letter to the third reviewer, authors described as

“In Ref. 21 (Page 361, Science 351, 361-365 (2016)), the authors described as follows: “To determine the active site conclusively, we develop four types of model catalysts with well-defined p-conjugation based on highly oriented pyrolytic graphite (HOPG): (i) pyridinic N-dominated HOPG (pyri-HOPG); (ii) graphitic N-dominated HOPG (grap-HOPG), and for comparison, (iii) edges patterned on the surface without N (edge-HOPG); and (iv) clean-HOPG (see supplementary methods and fig. S1).” Pyridinic N-dominated HOPG (pyri-HOPG) and graphitic N-dominated HOPG (grap-HOPG) mentioned by the authors don’t refer to pure pyridinic N or graphitic N doped HOPG. As shown in the inset of Fig. 1F in Ref. 21, the percentage of graphitic N in graphitic N-dominated HOPG is about 82.2 % (0.60/0.73) and the percentage of pyridinic N in pyridinic N-dominated HOPG is about 95 % (0.57/0.60).”

But the indicated figure extracted from (Science 351, 361-365 (2016)) in the reply letter by the authors show higher quality compared to the result by the authors in terms of selectivity of the peak component in the material. The derived percentage (pointed out by the authors) is just a result due to the fitting accuracy (I doubt poor fitting analysis by the authors now because there is no clear description about this important analysis). In any case, the general reader cannot convince that the XPS results in Figs. 3d-p are composed by a single component.

Moreover, there are other reports showing single component nitrogen doping (for example, J. Mate. Chem., 2011, 21, 8038).

I thus suggest that authors should revise the manuscript as originally the third referee pointed out.

Response:

Thanks for your valuable reminder. We refitted N 1s XPS peaks with the same width of FWHM around 1.9-2 eV, which further confirmed our conclusion that the single peak in N 1s XPS peaks corresponding to single pyrrolic nitrogen configuration of SPNCMs (F, 1/1), SPNCMs (F, 1/3), SPNCMs (F, 1/7), SPNCMs (Cl) and SPNCMs (Br). We also checked previous research ([37] Science 351, 361-365 (2016), [52] J. Mate. Chem. 21, 8038-8044 (2011) and [53] Carbon 125, 401-408 (2017)), single or high-purity pyridinic N doped carbon materials were indeed obtained in their works.

Thus, we revised our related description as “Up to now, there are few reports on single or high-purity pyridinic nitrogen-doped^{37,52,53} and graphitic nitrogen-doped carbon materials³⁷, and no reports on single pyrrolic nitrogen-doped carbon materials. Therefore, it is extremely desired to construct single nitrogen configuration-doped carbon materials, especially pyrrolic nitrogen doped

carbon materials.” in Line 11 Page 13.

Q2. Pyridonic nitrogen and hydrogenated pyridinic nitrogen

Authors added the description about pyridonic nitrogen to support the assignment for the pyrrolic nitrogen. The discussion is good. It is however not strong evidence but possible interpretation. The strong experimental evidences to support the assignment is still required, strictly to say.

Moreover, authors did not revise the manuscript for the hydrogenated pyridinic nitrogen though some descriptions are there in the reply letter.

Response:

Thanks for your valuable advice. In combination with the description “Protonated pyridinic-N (N-Q in low-rank coals), pyridine-N-oxide (N-X) and pyridone-N (part of N-5) are least stable and at relatively mild pyrolysis conditions (at 773 K or prepared in a DTF) they are converted to N-6, while pyrrolic-N (the other part of N-5) remains unaffected.” in the previous report [*Carbon* 33, 1021-1027 (1995)], XPS analyses of the SPNCMs (F, 1/1) samples after thermal treatment at 500, 550 and 600 °C, respectively, under nitrogen atmosphere for 1 h were conducted.

As shown in Figure R1a and b, high resolution XPS spectrum of N1 still shows the single peak around ~400.5 eV after thermal treatment temperature below 550 °C, but the pyridone-N will convert to pyridinic-N (binding energy of approximately 398.8 eV) at the same temperature according to ref. *Carbon* 33, 1021-1027 (1995). While the N 1s XPS spectrum of the SPNCMs (F, 1/1) samples treated under 600 °C for 1 h (Figure R1) shows two peaks at 398.8 eV and 401 eV, corresponding to pyridinic and quaternary nitrogen, respectively. Considering that instability of pyrrolic nitrogen higher than 600 °C, these results indicate the conversion of pyrrolic nitrogen to pyridinic and quaternary nitrogen. As mentioned above, we could reasonably conclude that single nitrogen configuration of the binding energy at approximately 400.5 eV for our nitrogen-doped carbon materials is NOT pyridone nitrogen.

As for hydrogenated pyridinic nitrogen, according to the referee’s previous advice “*Also, if authors show the drastic decrease of N1s peak by heating at higher temperature, authors may discuss that this is not come from hydrogenated pyridinic nitrogen (because pyrrolic nitrogen is known to be unstable against heating).*”, hydrogenated pyridinic nitrogen group and pyrrolic nitrogen group are both thermally unstable structures, but the thermal stability of hydrogenated pyridinic nitrogen group is even worse. As shown in Ref. Fig.1 [*Surface Science* 634, 89-94 (2015)], with the increase of temperature from 170 K to 650 K, the signals assigned to hydrogenated nitrogen decrease, and at 650 K the spectra (black line in Ref. Fig.1a) resemble those observed prior to hydrogenation (purple line in Ref. Fig.1e). We treated the SPNCMs (F, 1/1) samples at 500 and 550 °C (773 and 823 K) for 1 h. As shown in Figure R1a and b, high resolution N 1s XPS spectrum still shows the single peak around ~400.5 eV, indicated the nitrogen configuration is neither the incorporation of pyrrolic nitrogen and hydrogenated pyridinic nitrogen, nor the pure hydrogenated pyridinic nitrogen.

Based on the instability of pyrrolic nitrogen against heating, the results confirm the single nitrogen

configuration in our carbon materials is neither hydrogenated pyridinic nitrogen nor pyridone nitrogen configuration.

Figure R1. High-resolution N 1s XPS spectra of SPNCMs (F, 1/1) after thermal treatment at 500, 550 and 600 °C for 1 h, respectively.

Table R1. Binding energies and FWHM for SPNCMs (F, 1/1) treated at 500, 550, 600 °C, respectively.

Sample	Pyridinic N		Pyrrolic N		Quaternary N	
	B.E.(eV)	FWHM	B.E.(eV)	FWHM	B.E.(eV)	FWHM
SPNCMs (F, 1/1)-500			400.65	2		
SPNCMs (F, 1/1)-550			400.5	2		
SPNCMs (F, 1/1)-600	398.85	1.8			401	2

Ref. Fig. 1 a) Spectra in the N 1s region taken at 170 K at different hydrogen exposures, as denoted. Thermal evolution of NDG exposed to 180 L hydrogen: b) Deconvolution at 650 K

(highest recorded temperature); c) Color-coded density plot of the thermal evolution of H-NDG; spectra are collected ~ every 12 K; d) Spectra recorded at the lowest temperature (170 K); e) Spectra at characteristic temperatures. Quantitative analysis of all probed exposures: f) Graphitic nitrogen; g) pyridinic nitrogen, the symbols represent the experimental data, the lines serve as guide to the eye. [*Surface Science* 634, 89-94 (2015)]

Q3. “Significant figures” and instrumental calibration

In the revision, authors describe as “Therefore, interplanar spacing of the (002) plane of small microcrystalline graphite of SPNCMs (F, 1/1) was calculated to be 0.3645 nm, far larger than 0.3354 nm, interplanar spacing of the (002) plane of ideal graphite.”

But there is no description about the accuracy about the value 0.3645 nm. In the previous version of the manuscript, the value was very rough. The authors should clearly describe the reliable “Significant figures”. Also, authors should clearly describe how to calibrate the instrumental deviation of the measured distances from XRD, TEM and STEM from the exact value.

Response:

Thanks for the valuable reminder from the referee. The interplanar spacing of the (002) plane of small microcrystalline graphite of SPNCMs (F, 1/1) was calculated to be 0.3645 nm according to the Bragg’s Law R (1) as shown below:

$$2d\sin\theta=n\lambda \quad R (1)$$

Where d is interplanar spacing for the plane of sample; θ is angle between X-ray and corresponding plane; $n=1$; $\lambda_{Cu K\alpha}=0.15406$ nm. Thus, $d=[0.15406/2\sin(12.2)]$ nm \approx 0.3645 nm.

The X-ray diffractometer was calibrated by calibrating 2θ of the strongest peak (111) to 28.4° (JCPDS No.27-1402) with the RIGAKU Silicon-640 as standard sample before the measurements of our samples.

Meanwhile, due to their turbostratic structure and low crystallization degree, it is very difficult to precisely measure the representative interplanar spacing of our SPNCMs as one sp^2 -hybridized amorphous carbon from TEM image. Therefore, the value of the interplanar spacing from TEM images is no longer used for us.

Supplementary Figure 1. (a) XRD patterns of F-APF (1/1), NCMs (F,1/1) and SPNCMs (F,1/1); SEM (b), TEM (c), HRTEM (d), HAADF-STEM (e) images and the elemental mapping images

(carbon (f), nitrogen (g)) of SPNCMs (F,1/1).

Reviewer #3 (Remarks to the Author):

Specific comments:

Q1. First of all, the authors demonstrated the defects in the porous carbon mainly by Raman, for example: the author stated that “The Raman spectrum of NCMs (F, 1/1) has been added in Fig. 2a, and the corresponding ratio of intensity of D/G bands was determined to be 0.99, slightly lower than that of SPNCMs (F, 1/1) (1.03), possibly because alkaline activation induces more defective sites when nitrogen-containing functionals are completely converted to single pyrrolic nitrogen configuration. As shown in Figure 2a and Supplementary Figure 9, Raman spectra of SPNCMs (F, 1/1), SPNCMs (F, 1/3), SPNCMs (F, 1/7), and CMs (FPF) show similar feature of sp²-hybridized carbon materials with the disorder-induced D-band and a first-order graphite-like G-band. ID/IG values of these four carbon materials were determined to be 1.03, 0.96, 0.99, 1.02.” and “Similar to the Raman spectrum of NCMs (F, 1/1), that of SPNCMs (F, 1/1) (Fig. 2a) shows typical feature of sp²-hybridized carbon materials with abundant defects induced mainly by nitrogen doping, as inferred by the high intensity ratio of about 1.03 between the disorder-induced D-band at 1353 cm⁻¹ and a first-order graphite-like G-band at 1593 cm⁻¹ (ID/IG) of SPNCMs (F, 1/1)51.”

However, Raman D band does not in any sense whatsoever represent disorder carbons, though many people have published so. The authors should cite proper Raman papers on special Raman spectroscopy of graphene or carbon materials, not materials papers with often wrong interpretations of fundamental Raman results. Check work by Andrea Ferrari, Denis Basko, Mildred Dresselhaus, Ado Jorio, Rūchiro Saito, Kentaro Sato, or any other systematic work on Raman of graphene, defective graphene, amorphous carbon, defects and impurities in graphene-like materials.

The above interpretation is only acceptable for well-ordered graphene. In fact, it is the opposite for highly defective graphitic carbon. The Tunistra-Koenig (T-K) relation breaks down, and the explanation of high D/G peak corresponding to high defect is not often correct. In addition, assigning D/G ratio without a systematic fitting can be meaningless. A comprehensive and appropriate Raman analysis should be carried out to substantiate structural claims. What's more, it's better to present a statistical analysis of the Raman D/G peak ratios.

In this study, the D peak of the typical sample is too wide, and the diffraction peaks of (002) and (100) are broad, this kind of carbon just like amorphous carbon, rather than sp²-conjugated carbon! At least, the 2D band (the value should be labeled) in the Raman spectra and SAED should be provided for demonstrating the so-called sp²-conjugated structure!

The authors stated that “The ratio of intensity of D/G bands of NCMs (F, 1/1) was determined to be 0.99, slightly lower than that of SPNCMs (F, 1/1) (1.03), possibly because alkaline activation induces more defective sites,” However, they also described that Raman spectra of SPNCMs (F, 1/1), SPNCMs (F, 1/3), SPNCMs (F, 1/7), and CMs (FPF) with the ID/IG values

of 1.03, 0.96, 0.99, 1.02 show similar feature of sp^2 -hybridized carbon materials with the disorder-induced D-band and a first-order graphite-like G-band. It is inappropriate to interpret the data according to their subjective needs!

Response:

Thanks for your valuable advices. We learned so much Raman analysis understanding for carbon materials from the work of Andrea C. Ferrari, Denis M. Basko, Mildred S. Dresselhaus, Ado Jorio, Riichiro Saito, Kentaro Sato, and so on. Just as what the referee said, the high I_D/I_G intensity ratio corresponds to high defect involving sp^2 carbon sixfold ring is reasonable **for well-ordered graphene, and not suitable for nanocrystal graphite and amorphous carbon**. Considering that the relatively complex structure of carbon materials derived from phenol-formaldehyde resin are not well-ordered graphene and the absence of the finer characterization evidence to elucidate the specific meaning of I_D/I_G values for our four carbon materials up to now, high I_D/I_G intensity ratio ascribed to nitrogen doping-defects is not reliable.

Therefore, the description on quantitative analysis on the I_D/I_G intensity ratio has been deleted, and qualitative Raman analyses in our work are mainly used to demonstrate the formation of carbon materials containing sp^2 -hybridized carbon atoms by low temperature dehalogenation-induced and alkaline-activated pyrolysis of 3-halogenated phenol-3-aminophenol-formaldehyde (X-APF) co-condensed resin. As shown in Figure R2, Raman spectra of SPNCMs (F, 1/1), SPNCMs (F, 1/3), SPNCMs (F, 1/7), and CMs (FPF) show similar feature of broad and high D band involving breathing mode of sp^2 carbon six-atom rings at approximately 1353 cm^{-1} , G band involving the bond stretching of all pairs of sp^2 atoms at approximately 1593 cm^{-1} , and weak 2D band at approximately 2711 cm^{-1} , similar with the Raman spectra of nanographite consisting of sp^2 -conjugated carbon atoms (Ref. Fig. 2) [ACS Nano 4, 4206-4210 (2010)], consistent with amorphous carbon in the stage 2 from nanocrystalline graphite to sp^2 a-C^{54,55},

According to *Phil. Trans. R. Soc. Lond. A* 362, 2477-2512 (2004), there are three stages from graphite to tetrahedral amorphous carbon (ta-C), as depicted in Ref. Fig. 2:

- (1) graphite \rightarrow nanocrystalline graphite (nc-G),
- (2) nanocrystalline graphite \rightarrow sp^2 amorphous carbon (a-C),
- (3) a-C \rightarrow ta-C.

Ref. Fig. 2 Variation of the sp^2 configuration in the three amorphization stages. [ACS Nano 4, 4206-4210 (2010)]

Widen (002) and (100) diffraction peak of SPNCMs (F, 1/1) in XRD pattern (Supplementary Figure 1a) also confirm the amorphous carbon feature of SPNCMs (F, 1/1). To further confirm the existence for sp^2 -conjugated structure in SPNCMs (F, 1/1), Carbon K-edge EEL spectra (energy resolution 0.5 eV) measurement has been conducted for SPNCMs (F, 1/1). As shown in Figure R3,

the carbon K edge spectrum of SPNCMs (F, 1/1) and the inset shows a clear sp^2 signal with energy loss peaks at 283 eV ($1s \rightarrow \pi^*$) and 293 eV ($1s \rightarrow \sigma^*$), consistent with the feature of amorphous carbon with sp^2 -hybridized carbon atoms in previous report [*Journal of Electron Spectroscopy and Related Phenomena* 3, 232-236 (1974)]. Moreover, SPNCMs (F, 1/1), SPNCMs (F, 1/3), SPNCMs (F, 1/7), and CMs (FPF) as electrode materials could exhibit good capacitance and low potential drop, which means their good electrical conductivity ascribed to relatively high sp^2 -hybridized carbon content and sp^2 -conjugated carbon structure.

Thus, deletion of the calculations about I_D/I_G values of our final carbon materials, it does not affect our conclusions about the formation of single pyrrolic N-doped carbon materials.

Figure R2 Raman spectra of the samples including NCMs (F, 1/1), SPNCMs (F, 1/1) (a), SPNCMs (F, 1/3) (b), SPNCMs (F, 1/7) (c) and CMs (FPF) (d).

Ref. Figure 3. (A,B) Raman spectra of nanographite from a purified sample prepared at 875 °C

over MgO with a cyclohexane feedstock. Reaction time for A was 30 s and B was 5 min. (C) Raman spectrum from a purified sample prepared at 325 °C over MgO with cyclohexane as the feedstock and a reaction time of 10 min. (D) Schematic of a nanographene flake illustrating the large number of edge defects relative to the bulk nanographene sheet. The high number of edge defects relative to the honeycomb structure of the bulk flake leads to a high D band in the Raman spectra.[*ACS Nano* **4**, 4206-4210 (2010)]

Figure R3. Carbon K-edge EEL spectrum (energy resolution 0.5 eV) measured for SPNCMs (F, 1/1).

All the analyses mentioned above demonstrate that the bonding of SPNCMs (F, 1/1), SPNCMs (F, 1/3), SPNCMs (F, 1/7), and CMs (FPF) is still mainly sp^2 -hybridized. To describe our carbon materials with more accuracy, we have revised the term “ sp^2 -conjugated carbon” as “ sp^2 -hybridized carbon”

Ref. Fig. 4 K-shell ionisation-loss spectra of the three allotropes of carbon [*Journal of Electron Spectroscopy and Related Phenomena* 3, 232-236 (1974)].

Q2. Page 6, Line 19. “Additionally, EDS analysis shows uniform doping of nitrogen species into the turbostratic carbon framework for SPNCMs (F, 1/1) (Supplementary Fig. 1f, g).” Firstly, Because the elements of N and C are two neighbor light elements, it’s very hard to distinguish them from each other by this kind of element mapping, authors should reconsider it. Also , the shapes of the images in Supplementary Fig.s 1e, f, g are different!

Response:

Thanks for your valuable reminder. Energy dispersive X-ray spectroscopy (EDX) by STEM and SEM is a common supplementary means to analyze the element distribution of nitrogen-containing Carbon samples in combination with XPS data In Ref. [*J. Am. Chem. Soc.* 141, 482-487 (2019)] the authors described “Furthermore, energy-dispersive X-ray spectroscopy (EDX) analysis (Ref. Fig. 5) shows the uniform distribution of TF (thiourea-formaldehyde Polymer) component within the G3DTF”. In STEM image and the corresponding elemental mapping of PNCM were shown in Ref. Fig. 6d, “indicating the uniform distribution of nitrogen in the PNCM” [*Adv. Mater.* 29, 1702268 (2017)]. Thus, the EDX mapping was used to analyze the N element distribution of N-doped carbon materials to some extent.

In our work, the images (Supplementary Figure S1e, f, g) were used at their original states without any picture processing, such as brightness, contrast, except resolution and size which should meet the submission requirements. As shown in Supplementary Figure S1e, the thickness of gray boundary region is thinner than the bright white body area, indicating less material in boundary

region, which results in weaker signals of carbon (Supplementary Figure S1f) and nitrogen (Supplementary Figure 1g) elements in this boundary region than in the body area. Visually, objects of the same size with lighter colors are larger than with darker colors. When the brightness and contrast of STEM, carbon and nitrogen mapping images (Supplementary Figure S1 e, f, g) in photoshop software were simultaneously tuned up and down, respectively, these three images displayed the same shapes (Figure R4). Therefore, Supplementary Figure S1e, f, g indicates the same shapes of images in practice.

Ref. Fig. 5 Morphological, compositional, and electronic characterization. (a) SEM image of G3DTF; (b) Raman spectra of GO and G3DTF; (c) SEM image of G3DTF and EDX elemental mapping of carbon (C), oxygen (O), nitrogen (N), and sulfur (S). Scale bars correspond to 20 μm . [J. Am. Chem. Soc. 141, 482-487 (2019)]

Ref. Fig. 6 a) SEM image of the as-synthesized PNCM (insets are the optical images of the monolithic structure). b,c) TEM images of PNCM (inset of part (c) shows the HRTEM image). d) STEM image and the corresponding elemental mapping of PNCM. e) XRD patterns of three sets of PNCM samples. f) Ar physisorption isotherms of three sets of PNCM samples at 77 K. The physisorption volumes of PNCM-2 and PNCM-3 were increased by 200 and 400 $\text{cm}^3 \text{g}^{-1}$, respectively, for a clear demonstration. [Adv. Mater. 29, 1702268 (2017)]

Supplementary Figure 1. (a) XRD spectra of F-APF (1/1), NCMs (F,1/1) and SPNCMs (F,1/1); SEM (b), TEM (c), HRTEM (d), HAADF-STEM (e) images and the elemental mapping images (carbon (f), nitrogen (g)) of SPNCMs (F,1/1).

Figure R4. HAADF-STEM (a) images and the elemental mapping images (carbon (b), nitrogen (c)) of SPNCMs (F,1/1), the brightness increased by 40 % and contrast ratio reduced by 40 % in these images.

Q3. *The reviewer doesn't think the structures of NCMs and SPNCMs shown in Figure 1 are appropriate, as the O contents of the samples are not low!*

Response:

Thanks for your valuable advices. We report one kind of single nitrogen configuration-doped carbon materials, single pyrrolic N-doped carbon materials (SPNCMs) as ideal model materials, based on a facile engineering process, combined low temperature dehalogenation-induced and alkaline-activated pyrolysis of 3-halogenated phenol-3-aminophenol-formaldehyde (X-APF) co-condensed resin, considering that pyrolyzed dehalogenation-induced coupling reaction between adjacent benzene rings promotes enough graphitization of carbon materials at low temperature and subsequent alkaline-activated pyrolysis induces exclusively complete transformation of sp^3 -hybridized nitrogen into sp^2 -hybridized pyrrolic nitrogen species. Therefore, Fig.1 in our work aims to present the complete conversion of the initial nitrogen configurations to the final single pyrrolic nitrogen configuration through a simple and clear scheme in the synthetic procedures of SPNCMs, highly consistent with our key idea in the manuscript. If the conversions for oxygen configurations are added in the structure, it will possibly make the readers confused for the formation of SPNCMs for the readers.

Q4. In the Reviewer's opinion, the typical sample in this study is N and O co-doped carbon, however, the authors didn't take the effect of O on the dopant N into account at all!

Response:

Thanks for the valuable reminder from the referee. As mentioned by the referee, there are indeed the interaction among all the atoms in carbon materials. In our SPNCMs, nitrogen species exists in single pyrrolic nitrogen configuration without oxidized nitrogen configuration. In other words, there is no direct N-O binding in our SPNCMs. Unfortunately, it is hard to investigate the effect of O on the dopant N between non-binding oxygen and nitrogen atoms in carbon materials due to the absence of reliable characterization means based on current technology. Next technologic revolutions are expected to allow us to explore this effect in the future. We shall investigate the effect of O on the dopant N as valued study if enough technologic demands are met.

Q5. The authors stated that "thermal treatment below about 600 °C always yields lowly graphitized nitrogen-doped carbon materials with poor electrical conductivity..." What are the specific values of conductivity of the samples in this study, how about these values compared with commercial conductive carbon?

Response:

Thanks for your valuable reminder. The exact value of the conductivity require large amount of sample, unfortunately, the research activity in the whole world is deeply affected, including us, due to the spread of COVID-2019, it's difficult to achieve this measurement.

Fortunately, the difference between galvanostatic charge-discharge curves of our single pyrrolic nitrogen-doped carbon materials (SPNCMs(F, 1/1), (SPNCMs(F, 1/3) and (SPNCMs(F, 1/7)) and that of control sample, nitrogen-doped carbon materials derived from 3-aminophenol-formaldehyde resin (NCMs (APF)) gives related information about the conductivity of the samples in the potential window of 1 V at the current density of 1A/g in three-electrode system in 1.0 M H₂SO₄ electrolyte. As shown in Figure R5 the potential drops of SPNCMs (F, 1/1), (SPNCMs (F, 1/3), (SPNCMs (F, 1/7) and NCMs (APF) were determined to be 45, 53, 50 and 640 mV, respectively. Considering that the potential drops of carbon materials is negatively dependent on the electrical conductivity of the samples under the same other condition such as sample quality and electrochemical tests, the conductivities of our single pyrrolic nitrogen-doped carbon materials (SPNCMs(F, 1/1), (SPNCMs(F, 1/3) and (SPNCMs(F, 1/7) is far higher than NCMs (APF) using no 3-fluorophenol. Relatively high potential drop of 640 mV shows NCMs (APF) after 3-aminophenol-formaldehyde resin pyrolyzed at 500 °C and activated at 500 °C with relatively low electrical conductivity, similar with previous report that electrical conductivity of pyrolyzed phenol-formaldehyde resin at 500 °C is down to 1.07×10⁻¹⁰ S cm⁻¹ at 298 K (Ref. Fig. 7, Ref. Table 1 and Table2) [*Journal of Non-Crystalline Solids* 12, 115-128 (1973)]. Their conductivities follow the equation:

$$\sigma = A \cdot e^{-B/T^{1/4}} \quad R(2)$$

where σ is the conductivity, Coefficients A and B for equation R(2) are listed in Ref. Table 1. The values of σ at 298 K (T) for carbon materials pyrolyzed under different temperatures were calculated by the equation R(2) and listed in Ref. Table 2. The results mentioned above confirm that our low temperature dehalogenation-induced and alkaline-activated pyrolysis strategy of

3-halogenated phenol-3-aminophenol-formaldehyde (X-APF) co-condensed resin considerably enhances the electrical conductivity of our SPNCMs to meet the demand of electrochemical applications for the conductivity of the carbon materials. In other words, the conductivities of our SPNCMs could be comparable to pyrolyzed phenol resin at far higher temperature than 500 °C. Considering that the conductivity of pyrolyzed phenol resin at 1200 °C ($2.22 \times 10^2 \text{ S cm}^{-1}$, Ref. Table 2 [*Journal of Non-Crystalline Solids* 12, 115-128 (1973)]) is still lower than that of graphite (10^3 S cm^{-1}) (Ref. Table 3) [*Powder Technology* 221, 351-358 (2012)], we conclude that the conductivities of our SPNCMs is lower than that of graphite. Although we still do not compare the conductivities of our SPNCMs with that of Carbon Black (10 S cm^{-1}) (Ref. Table 3) [*Powder Technology* 221, 351-358 (2012)], obviously, the conductivities of our SPNCMs have meet the demand of electrochemical application.

Figure R5. Galvanostatic charge/discharge (GCD) curves of NCMs (APF), SPNCMs (F, 1/1), SPNCMs (F, 1/3), SPNCMs (F, 1/7) and CMs (FPF) as work electrodes at a current density of 1 A g^{-1} in three-electrode system in $1.0 \text{ M H}_2\text{SO}_4$ electrolyte, respectively.

Ref. Fig. 7 Dc conductivity of thermally degraded phenol formaldehyde resins versus $T^{-1/4}$.
 [Journal of Non-Crystalline Solids 12, 115-128 (1973)]

Ref. Table 1 Coefficients A and B for equation R(2) (pyrolyzed phenol formaldehyde resins).
 [Journal of Non-Crystalline Solids 12, 115-128 (1973)]

HTT (°C)	A ($\Omega \cdot \text{cm}$) ⁻¹	B (deg K ^{1/4})
400	5.1×10^{28}	448
450	9.1×10^{21}	339
500	8.1×10^{14}	238
550	3.6×10^7	115
600	3.4×10^4	57.1
650	3.3×10^3	29.1
700	1.1×10^3	18
750	4.4×10^2	10.4
800	3.4×10^2	6.26
850	2.9×10^2	4.01
900	2.9×10^2	2.91
1000	3.0×10^2	1.97
1100	3.2×10^2	1.78
1200	3.3×10^2	1.65

Ref. Table 2 Coefficients A and B for equation R(2) (pyrolyzed phenol formaldehyde resins), and
 the calculated values of σ at 298 K (T).

HTT (°C)	A ($\Omega \cdot \text{cm}$) ⁻¹	B (deg K ^{1/4})	σ ($\Omega \cdot \text{cm}$) ⁻¹
400	5.1×10^{28}	448	7.57×10^{-19}
450	9.1×10^{21}	339	3.34×10^{-14}

500	8.1×10^{14}	238	1.07×10^{-10}
550	3.6×10^7	115	3.43×10^{-5}
600	3.4×10^4	57.1	3.66×10^{-2}
650	3.3×10^3	29.1	3.00
700	1.1×10^3	18	14.45
750	4.4×10^2	10.4	36.01
800	3.4×10^2	6.26	75.36
850	2.9×10^2	4.01	1.10×10^2
900	2.9×10^2	2.91	1.44×10^2
1000	3.0×10^2	1.97	1.87×10^2
1100	3.2×10^2	1.78	2.08×10^2
1200	3.3×10^2	1.65	2.22×10^2

Ref. Table 3 Material and compact characteristics. [*Powder Technology* 221, 351-358 (2012)]

Filler	BET surface area (m ² /g)	Conductivity (S/m)		Isolated Single particle conductivity	Filler contribution limit from compact ^a
		Powder compact at 5 MPa	Paper		
MWCNTs	272	5.43×10^2	5×10^3	$10^6 - 10^{7b}$	10.3×10^3
Graphene	180	2.62×10^2	1.4×10^3	$10^7 - 10^8$ [2]	10.9×10^3
Carbon Black	56.9	5.58×10^2	9×10^1	10^3 [14]	8.8×10^3
Graphite	3.08	2.12×10^3	1.2×10^3	10^5 [27] ^c	13.8×10^3

^a Estimate based on the model as described in [17].

^b As provided by the Nanocyl Company: <http://www.nanocyl.com/en/CNT-Expertise-Centre/Carbon-Nanotubes>.

^c Value highly variable depending on source.

Q6. The topic of this work is “sp²-conjugated Carbon Materials”, why the authors cited 16 of Ref.s on sp³ and sp carbon in the introduction section in the revised version? So much Ref.s (1/4 of the total) make the reviewer a little confused!

Response:

Thanks for your valuable reminder. The introduction of N-doped sp²-hybridized carbon materials in the initial manuscript seemed to be just a very small area among the research on carbon materials. The properties and applications of carbon-based materials are strongly associated with the hybridization of the carbon atoms including sp³, sp² and sp, and doping heterogenous atoms, including nonmetal atoms (e.g. B, N, O, S, P) and metal atoms (e.g. Pt, Fe, Co, Ni, Zn, Mo), into the framework of carbon-based materials. With the purpose of providing more complete overview of the field of carbon materials, we have cited the corresponding typical research works in the revised version. We also expect that this research could be expand to more related carbon-based materials.

Q7. For HR-XPS spectra, the basic rules of the deconvolution should be obeyed. At least, the half-width of each peak for one element should be consistent! There are still some problems of the deconvolution in Figure 3f, Supplementary Figure 6c, and Supplementary Figure 12c in the revised version!

Response:

Thanks for the valuable advice from the referee. We refitted XPS peaks with the same width of FWHM approximately 1.9 eV for N 1s and approximately 1.8 eV for O 1s of the samples, and these changes don't affect our conclusions. All the binding energies and FWHMs for N1s and O 1s for the samples have been listed below:

Supplementary Table 2. Binding energies and FWHM for N1s in Figure 2 and Figure 3

Sample	-NH-, >N-		Pyrolic N		Figure
	B.E.(eV)	FWHM	B.E.(eV)	FWHM	
F-APF (1/1)	399.25	1.9	-	-	Figure 2c
NCMs (F, 1/1)	398.9	1.9	400.3	2	Figure 2d
SPNCMs (F, 1/1)	-	-	400.25	2	Figure 2e
F-APF (1/3)	399.53	1.9	-	-	Figure 3b
NCMs (F, 1/3)	398.97	1.9	400.57	2	Figure 3c
SPNCMs (F, 1/3)	-	-	400.25	2	Figure 3d
F-APF (1/7)	399.55	1.9	-	-	Figure 3f
NCMs (F, 1/7)	399	1.9	400.4	2	Figure 3g
SPNCMs (F, 1/7)	-	-	400.35	2	Figure 3h
Cl-APF	399.5	2	-	-	Figure 3j
NCMs (Cl)	398.9	1.9	400.3	2	Figure 3k
SPNCMs (Cl)	-	-	400.35	2	Figure 3l
Br-APF	399.5	1.9	-	-	Figure 3n
NCMs (Br)	398.9	1.9	400.5	2	Figure 3o
SPNCMs (Br)	-	-	400.4	2	Figure 3p

Supplementary Table 3. Binding energies and FWHM for N1s of NCMs (APF) and NCMs (F, 3/1) in Supplementary Figure 3

Sample				
	B.E.(eV)	FWHM	B.E.(eV)	FWHM
NCMs (APF)	400	1.9		
NCMs (F, 3/1)			401.7	2

Supplementary Table 4. Binding energies and FWHM for N1s and O1s of 3-fluorophenol-2-hydroxypyridine-formaldehyde resin, F-HPF (F, 1/1) in Supplementary Figure 4

Sample	Pyridinic N		Pyridonic N		-C=O		-O-H, C-O-C	
	B.E.(eV)	FWHM	B.E.(eV)	FWHM	B.E.(eV)	FWHM	B.E.(eV)	FWHM
F-HPF	398.95	1.9	400.35	1.9	531.3	1.8	532.95	1.8

(F, 1/1)

Supplementary Table 5. Binding energies and FWHM for O1s in Supplementary Figure 6

Sample	-C=O		-OH, C-O-C	
	B.E.(eV)	FWHM	B.E.(eV)	FWHM
SPNCMs (F, 1/1)	532	1.85	533.5	1.9
SPNCMs (F, 1/3)	531.9	1.8	533.3	1.8
SPNCMs (F, 1/7)	532.8	1.85	533.7	1.9
CMs (FPF)	532.1	1.8	533.3	1.8

Supplementary Table 6. Binding energies and FWHM for N1s and O1s of SPNCMs (F, 1/1) in Supplementary Figure 12

Sample	Pyrrolic N		Electrochemically oxidized pyrrolic N		-C=O		-O-H, C-O-C	
	B.E.(eV)	FWHM	B.E.(eV)	FWHM	B.E.(eV)	FWHM	B.E.(eV)	FWHM
After charging	400.2	2	402	2	532.1	1.8	533.3	1.8
After discharging	400.2	2	401.8	2	532.2	1.8	533.2	1.85

Supplementary Table 7. Binding energies and FWHM for N1s and O1s of SPNCMs (F, 1/1) in Supplementary Figure 13

Sample	Pyrrolic N		Electrochemically oxidized pyrrolic N		-C=O		-O-H, C-O-C	
	B.E.(eV)	FWHM	B.E.(eV)	FWHM	B.E.(eV)	FWHM	B.E.(eV)	FWHM
After 30,000 cycles	400.4	2	401.8	2	532	1.8	533.25	1.8

Reviewers' Comments:

Reviewer #2:

Remarks to the Author:

I have carefully checked the revision parts. The authors well revised the manuscript. Now, the quality of the manuscript becomes much better than previous version. General reader can be convinced the "Single Pyrrolic Nitrogen Species" as well as its great pseudocapacitance. Thus I recommend the manuscript to be published.

Reviewer #3:

Remarks to the Author:

The authors have done a lot of work to address the reviewers' concerns, and the quality of the manuscript has been improved. Although the reviewer understand the technical limitation and the impact of the COVID-2019, the reviewer still can't be persuaded by the present data. Reviewer doesn't think the structure of the SPNCMs is as what they have shown in Figure 1c. By the XPS, XRD, and Raman analysis, the SPNCMs should have abundant defects and a certain amount of oxygen. Therefore, the reviewer really can not accept it for publication unless the authors provide more direct evidences for this structure (Figure 1c), although the author cites a lot of literature.

Reviewers' comments:

Reviewer #2 (Remarks to the Author):

I have carefully checked the revision parts. The authors well revised the manuscript. Now, the quality of the manuscript becomes much better than previous version. General reader can be convinced the "Single Pyrrolic Nitrogen Species" as well as its great pseudocapacitance. Thus I recommend the manuscript to be published.

Response:

Thanks for your careful review for our manuscript. Those valuable advices from the reviewer has helped us to promote the quality of our manuscript.

Reviewer #3 (Remarks to the Author):

The authors have done a lot of work to address the reviewers' concerns, and the quality of the manuscript has been improved. Although the reviewer understand the technical limitation and the impact of the COVID-2019, the reviewer still can't be persuaded by the present data. Reviewer doesn't think the structure of the SPNCMs is as what they have shown in Figure 1c. By the XPS, XRD, and Raman analysis, the SPNCMs should have abundant defects and a certain amount of oxygen. Therefore, the reviewer really can not accept it for publication unless the authors provide more direct evidences for this structure (Figure 1c), although the author cites a lot of literature.

Response:

Thanks for your careful review from the referee. As mentioned by the referee, by the XPS, XRD, and Raman analysis, the SPNCMs indeed have abundant defects and a certain amount of oxygen.

Generally, KOH activation could generate nanopores, leading to an increase in defect density for carbon materials. Up to date, it is extremely difficult to precisely control and determine the defect type and density of defects for carbon materials due to the limitations of present characterization. Fortunately, specific surface area represents the density of nanopore-dependent defects for our final carbon materials to a large content. Moreover, the results of electrochemical surface area (ESA) tests in the non-faradaic voltage range is completely consistent with specific surface areas for SPNCMs (F, 1/1), SPNCMs (F, 1/3), SPNCMs (F, 1/7) and CMs (F, 1/1). Therefore, the capacitance from these defects could be approximately regarded as the double layer capacitance. Related discussion has been added in this revised version.

We have been revised the presentative structure of the SPNCMs (Figure 1c), and possible oxygen-containing groups including hydroxyl and carbonyl groups have been introduced into the structure of the SPNCMs according to High-resolution O 1s XPS analysis of the SPNCMs (Supplementary Figure 6).

Fig. 1 Schematic of synthetic procedure of SPNCMs. The synthetic procedure of SPNCMs based on a strategy of low-temperature dehalogenation-induced and subsequent alkaline-activated pyrolysis of 3-halogenated phenol-3-aminophenol-formaldehyde (X-APF) co-condensed resins.

Supplementary Figure 6. High-resolution O 1s XPS spectra of SPNCMs (F, 1/1) (a), SPNCMs (F, 1/3) (b), SPNCMs (F, 1/7) (c), and CMs (FPF) (d).